# Latent brain subtypes of chronotype reveal unique behavioral and health profiles across population cohorts

Le Zhou [1,2], Karin Saltoun [1,2], Justin Marotta[1,2], Shambhavi Aggarwal[1,2], Jakub Kopal [3], Julie Carrier [4,5], Kai-Florian Storch [6,7], Robin I. M. Dunbar [8] & Danilo Bzdok [1,2] ✉

Chronotype is shaped by the complex interplay of endogenous and exogenous factors. This time-enduring trait ties into societal behaviors and is linked to psychiatric and metabolic conditions. Despite its multifaceted nature, prior research has treated chronotype as a monolithic trait across the population, risking overlooking substantial heterogeneity in neural and behavioral fingerprints. To uncover hidden subgroups, we develop a supervised pattern-learning framework integrating three complementary brain-imaging modalities with deep behavioral and health profiling from 27,030 UK Biobank participants. We identify five distinct, biologically valid chronotype subtypes. Each demonstrates unique patterns across brain, behavioral and health profiles. External validation in 10,550 US children from the ABCD Study cohort reveals reversed age distributions and replicates sex-associated brain-behavioral patterns, suggesting that potential divergences between chronotype traits observed throughout adulthood may begin to emerge early in life. These findings highlight underappreciated sources of population variation that echo the rhythm of people's inner clock.

Over the course of human evolution, variation in sleep-wake cycles probably emerged as an adaptive strategy to ensure group survival and safety. Such temporal specialization allowed for complementary vigilance periods, with morning-oriented people facilitating orderly daytime activities, while evening-oriented individuals maintained protective watchfulness during nighttime hours[1,2]. In modern society, this divergence still manifests in various societal roles, such as early healthcare workers and night security personnel. These evolutionarily conserved sleep-wake phenotypes reflect underlying chronotype variations, which refer to the diurnal preferences in sleep and alertness timing across the 24-h cycle[3,4].

As a multifaceted construct, inter-individual differences in chronotype have been linked to various consequences in recent research, including physical and mental health, lifestyle choices, work efficiency, and societal functioning[5-9]. With the rise of irregular and variable lifestyles in the digital era and, later, post-covid era, sleep patterns have become increasingly diverse and complex[10-15]. Traditional dichotomic views, morning versus evening types, may not fully capture this behavior per se or its associated implications in contemporary society. A sufficiently nuanced reconceptualization of the chronotype is needed to help modern humans optimize work-life balance, reduce associated health

[1]TheNeuro - Montreal Neurological Institute (MNI), Department of Biomedical Engineering, McGill University, Montreal, QC, Canada. [2]Mila - Quebec Artificial Intelligence Institute, Montreal, QC, Canada. [3]Centre for Precision Psychiatry, Division of Mental Health and Addiction, Oslo University Hospital & Institute of Clinical Medicine, University of Oslo, Oslo, Norway. [4]Department of Psychology, Universite de Montreal, Montreal, QC, Canada. [5]Center for Advanced Research in Sleep Medicine, Research center of the Centre integre universitaire de santé et de services sociaux du Nord de l'Ile-de-Montreal, Montreal, QC, Canada. [6]Department of Psychiatry, McGill University, Montreal, QC, Canada. [7]Douglas Mental Health University Institute, McGill University, Montreal, QC, Canada. [8]Department of Experimental Psychology, University of Oxford, Oxford, UK. ✉e-mail: danilo.bzdok@mcgill.ca

vulnerabilities, and enhance well-being across age groups and societal sectors.

In fact, an unexpectedly broad variety of chronotype phenotypes may arise from a complex interplay of exogenous and endogenous factors. Light exposure serves as the primary environmental cue for entrainment, directly synchronizing the biological clock through retinal input[3]. Social demands, such as work schedules and societal roles, constitute another critical exogenous factor that can sculpt people's everyday sleep-wake cycles[16]. On the endogenous side, gene variants and hormonal fluctuations play key roles in modulating individual sleep-wake preferences[17-19]. In fact, sleep-wake preferences typically follow a U-shaped pattern through the lifespan, with morning preference in early childhood shifting later during adolescence before returning to earlier wake-up times again with aging[20-22]. Further, sex differences have been reported in how chronotype is distributed in society, with women generally showing stronger morning preferences than men, although these distinctions may fade around the onset of menopause[22-26].

A deeper understanding of individual variation in sleep-wake cycles and their associations with health outcomes is becoming increasingly important. This is because numerous studies have linked the late chronotype to heightened health risks, including psychiatric and metabolic sequelae compared to the early type[27-31]. Recent advances in neuroscience have enabled researchers to explore brain structure and function in greater depth, revealing links between chronotype and brain circuits including limbic system, basal ganglia, and prefrontal cortex. These reports suggested potential neural manifestations of chronotype which may explain its associations with emotional regulation, reward processing, as well as cognitive function[9,31-37]. However, the existing literature has often conceptualized early-riser and night-owl groups as phenotypically uniform, potentially neglecting the substantial heterogeneity in neural and behavioral profiles that may coexist within the same chronotype. For example, a recent psychiatric meta-analysis reported substantial variation in the association between chronotype and depression[38]. Moreover, to the best of our knowledge, there is limited research that has examined chronotype in relation to neurobiology, detailed behavioral phenotypes, and health outcomes simultaneously across a broader range of age groups. Collectively, neuroscience could substantially benefit from a more holistic investigation covering the individual differences in chronotype and its association to a diverse spectrum of phenotypes and health outcomes across age groups at population scale.

In the present study, we developed a supervised pattern learning framework that integrated three complementary brain imaging measurements - gray matter volume, integrity of major white matter tracts, and intrinsic functional connectivity - with 977 behavioral phenotypes as well as ~1500 diagnostic assessments, and over 6000 medications from 27,030 UK Biobank participants to investigate chronotype-associated heterogeneity. Our large-scale, multimodal approach was able to reveal five distinct brain-defined biological subtypes of chronotype, each exhibiting unique behavioral and clinical profiles. For the purpose of external validation, we carried over our extracted brain-behavior patterns to a younger age cohort, deeply profiled with 5,859 curated phenotypes from 10,550 children in the USA-based ABCD Study® cohort. Our collective findings demonstrate that chronotype-brain relationships are far more heterogeneous in the wider population than previously recognized, advancing our understanding of chronotype variability, and may inform targeted chronotherapeutic interventions.

## Results
### Five distinct subtypes emerge from pattern learning of brain and chronotype
Our study was motivated by the hypothesis that the chronotype may associate with the brain and behavior in diverse, and potentially distinct, ways across individuals. To that end, we applied partial least squares (PLS) as the pattern-learning algorithm to analyze the binarized chronotype and three brain imaging modalities: regional gray matter volume, white matter integrity of major tracts, and resting-state functional connectivity. After 1000 empirical permutation iterations, the latent factor correlation of the leading 5 components was found to be statistically significant at the threshold for component-wise comparison ($P < 0.001$). For each participant, we derived 5 brain scores, reflecting the degree to which their neurobiological profiles aligned with each of the 5 identified patterns. Comparing brain scores between night owls and early birds, as defined by self-reported chronotype, revealed three modes subdividing night owls and two modes subdividing early birds. To assess the relative importance of each brain feature (brain loadings) for the 5 identified brain patterns, we performed bootstrapping analysis with 1,000 iterations to evaluate the robustness of brain features (SFig 1-5). Overall, despite starting out from a categorical UK Biobank indicator variable, our approach identified 5 distinct brain-chronotype modes, comprising three eveningness modes and two morningness subtypes in the wider UK population.

### Subtype 1: a night owl pattern associated with emotional regulation and cognitive performance
The first discovered subtype represented a night owl pattern, characterized by significantly higher brain scores in night owls than in early birds (two-sample $t$-test for morningness >eveningness: $t(27,028) = -16.94$, $P < 0.001$), indicating greater "eveningness". Strong signals in regional gray matter volume (GMV) variations were primarily observed in the limbic system, frontal brain regions, and basal ganglia (Fig. 1a, Supplementary Fig. 1a). Specifically, positive loadings in the paracingulate gyrus, amygdala, parahippocampal gyrus, frontal orbital cortex (OFC), frontal pole, inferior frontal gyrus, frontal medial cortex, superior frontal gyrus, superior temporal gyrus, and temporal lobe, indicating that increased GMV in these areas was associated with expression of this night owl pattern. In contrast, we observed negative loadings in the caudate, pallidum, and cingulate gyrus, suggesting that reduced GMV in these areas also contributed to this subtype's expression in UKB participants. Similarly, white matter integrity measures showed uniformly positive loadings, particularly in the left anterior corona radiata, fornix, and tapetum, indicating that stronger integrity in these tracts was associated with this eveningness subtype (Fig. 1b, Supplementary Fig. 1b). At the functional connectivity level (Fig. 1c, Supplementary Fig 1c), positive loadings were identified for connections within the primary somatosensory cortex, primary motor cortex, and somatomotor medial cortex, whereas their links to the basal ganglia and cerebellum had negative loadings. In limbic networks, connections between the temporal cortex and somatosensory cortex had positive loadings, while those between the cingulate cortex and somatosensory cortex had negative loadings. Additionally, the connection strength between the frontoparietal network and attention networks was positively weighted, contributing to this eveningness pattern.

To further characterize this first subtype in phenome-wide profiling at population scale, we extracted its brain scores for each participant and conducted three large-scale association assays: a phenome-wide association study (PheWAS) based on 977 behavioral indicators, a diagnosis-wide association study (DiaWAS) using 1,396 diagnostic records, and a medication-wide association study (MedWAS) covering 133 Anatomical Therapeutic Chemical (ATC) Classification System categories at the pharmacological mechanism level (level 3) (Fig. 1d-f, Supplementary Table 2). The results revealed that subtype 1 is associated with lifestyle factors such as living with children, fast driving, mobile phone usage, alcohol consumption, and smoking. In the cognitive domain, it was linked to faster reaction times and better performance in puzzle-solving and digit calculation tasks. Within the mental health category, this mode was linked with irritability/mania-

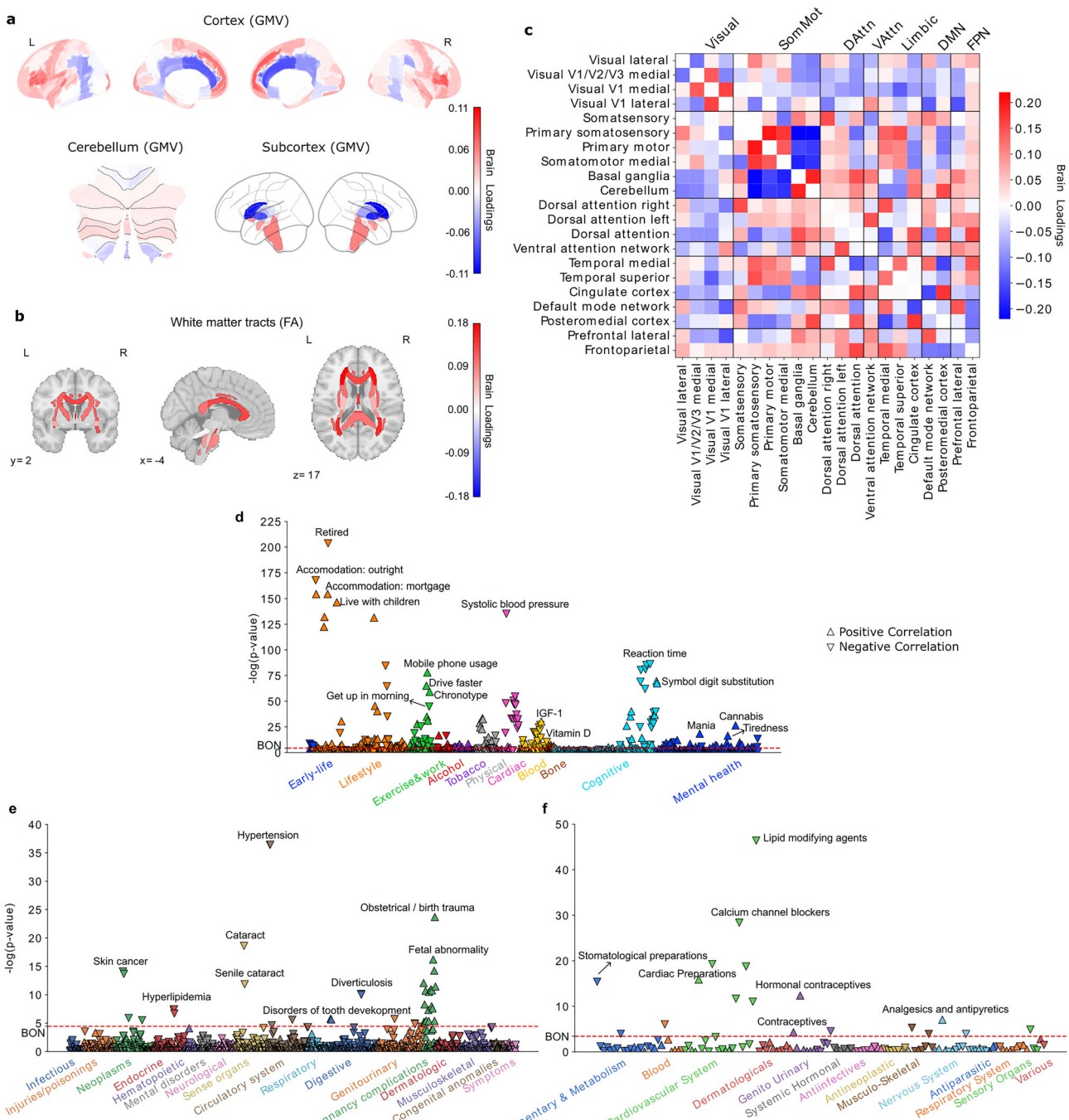

**Fig. 1 | Subtype 1 is a night owl pattern associated with emotional regulation, fast reaction time, as well as increased white-matter integrity. a** Brain loadings of gray matter volume (GMV) across cortical, subcortical, and cerebellar regions, highlighting the basal ganglia, limbic cortex, amygdala, and frontal cortex. **b** Brain loadings of fractional anisotropy (FA) for white matter microstructures. **c** Brain loadings of functional connectivity between 21 meaningful ICA components. Abbreviations: Visual: Visual Network; SomMot: Somatomotor Network; DAttn: Dorsal Attention Network; VAttn: Ventral Attention Network; Limbic: Limbic Network; DMN: Default Mode Network; FPN: Frontoparietal Network. **d–f** The phenome-wide association study with behaviors, diagnoses and medications. Pearson's correlation results for each phenotype are presented on a logarithmic scale of associated P-values. The horizontal line denotes the significance thresholds following Bonferroni correction (*P* < 0.05). Abbreviations: IGF-1: Insulin-like growth factor 1. Full details are provided in Supplementary Table 2. Source data are provided as a Source Data file.

related phenotypes, tiredness, feelings of being fed up, risk-taking behavior, and taking cannabis. Notably, in the blood assay tests, Vitamin D levels were negatively associated with this mode, suggesting potential reduced sunlight exposure among individuals with this night owl pattern. In DiaWAS, this mode was positively correlated with pregnancy complications, including birth trauma, fetal distress, abnormal labor forces, and miscarriage (but stillbirth), aligning with

MedWAS findings in the genitourinary system and sex hormones category (e.g., systemic and topical hormonal contraceptives). Additionally, this mode was positively associated with medications related to cardiac function (C01E - Other Cardiac Preparations) and pain relief (N02B - Other Analgesics and Antipyretics). Overall, this night owl pattern was associated with risky lifestyle behaviors, difficulties in emotional regulation, and better cognitive performance.

## Subtype 2: a night owl pattern linked to depression, smoking and cardiovascular risks

The second most important subtype also represented a night owl pattern, with its brain scores significantly higher in the eveningness group (t(27,028) = -15.81, $P < 0.001$). The dominant feature of brain variations in this mode located to white matter integrity, which was uniformly negatively associated with chronotype, suggesting that reduced integrity of white matter tracts was a defining feature of this night owl subtype (Fig. 2b, Supplementary Fig. 2b). Moreover, we primarily observed GMV variations in the limbic system, frontal areas, and basal ganglia (Fig. 2a, Supplementary Fig 2a). Specifically, positive loadings in the caudate, thalamus, insular, frontal regions indicated that increased GMV in these regions contributed to this subtype's expression in UKB participants. Conversely, negative loadings in cerebellum regions suggested that reduced GMV in these regions was also associated with this subtype. Functional connectivity variations were centered on the somatosensory cortex, temporal cortex, and attention networks (Fig. 2c, Supplementary Fig. 2c). Connections between somatosensory cortex and temporal networks showed negative loadings, whereas links between the temporal cortex and the basal ganglia, cerebellum, and left dorsal attention networks showed positive loadings. Overall, the brain pattern of this mode highlighted the reduced integrity of white matter tracts, and key implications of basal ganglia, and limbic system in characterizing this night owl subtype.

Our PheWAS assay (Fig. 2d-f, Supplementary Table 2) revealed that subtype 2 is linked with smoking and playing computer games as well as late wake-up times, lower income, lower participation in sports clubs, and less vigorous physical activity in lifestyle-related categories. Additionally, this mode was related to high blood pressure in PheWAS, consistent with hypertension and cerebrovascular disease in DiaWAS, as well as blood and cardiovascular-related medications in MedWAS (e.g., B01A – Antithrombotic Agents, C08C – Selective Calcium Channel Blockers with mainly Vascular Effects, C09A – ACE Inhibitors, C10A – Lipid Modifying Agents) – showing convergence of findings across three separate analyses and data sources. Moreover, its association with blood glucose levels in PheWAS aligned with diabetes in DiaWAS and the use of glucose-lowering drugs in MedWAS. In the mental health category of PheWAS, depression-related phenotypes were prominent, aligning with antidepressant medications in MedWAS. Furthermore, consistent with the association with smoking, this mode was also linked to chronic bronchitis and pneumonia in DiaWAS. In summary, across large-scale phenotype assays at phenome scale, subtype 2 co-occurred with smoking, depression, and variables related to cardiovascular risks or disease in the UK Biobank cohort.

## Subtype 3: a morningness pattern with less substance use and fewer health issues

The next brain-extracted subtype emerged as an early bird pattern, with its brain scores significantly higher in the morningness group (t(27,028) = 8.44, $P < 0.001$), indicating that higher expression of this subtype corresponds to an earlier chronotype. Variations in the brain features for this mode showed some similarity to those observed in the eveningness-associated subtype 1 in GMV (Pearson's $r = 0.75$, $P < 0.001$) and FA (Pearson's $r = 0.623$, $P < 0.001$), but opposite in functional couplings (Pearson's $r = -0.429$, $P < 0.001$) (Supplementary Fig. 6). Specifically, positive loadings in the frontal lobe areas (e.g., frontal medial cortex, frontal pole, frontal operculum cortex, inferior frontal gyrus, superior frontal gyrus), parahippocampal gyrus, paracingulate gyrus indicated that greater GMV in these regions contributed to stronger this early bird expression. In contrast, negative loadings in caudate, pallidum and cingulate gyrus suggested that lower GMV in these areas is associated with this early bird-like subtype (Fig. 3a, Supplementary Fig. 3a). Similarly, positive loadings in fornix fiber bundle, tapetum, and superior fronto-occipital fasciculus indicated that greater integrity of these white matter tracts contributed to

this subtype's expression (Fig. 3b, Supplementary Fig. 3b). At the functional connectivity, patterns within the somatosensory and motor networks were opposite to those in subtype 1, with the connections between primary somatosensory cortex and motor cortex being negatively correlated with this subtype's expression while their links to basal ganglia and cerebellum were positively correlated (Fig. 3c, Supplementary Fig. 3c). Notably, connections related to limbic networks in this subtype, including somatosensory networks, dorsal attention network, ventral attention network, default mode network, and frontoparietal network, were largely shown opposite directions to subtype 1. For example, we observed positive loadings in the couplings between basal ganglia and temporal cortex in subtype 3, while these connections showed negative loadings in subtype 1. Overall, the functional coupling patterns supporting the expression of subtype 3 exhibited significant differences in the opposite directions compared to subtype 1, particularly in basal ganglia, limbic networks, and default mode network, reflecting a distinct early bird connectivity profile.

Regarding phenome-wide profiling at population scale (Figs. 3d–f, Supplementary Table 2), the PheWAS results revealed that this mode is associated with getting up earlier, confirming our main analysis, spending more time on TV, consuming less alcohol, and having never smoked in lifestyle-related categories. In terms of mental health reports, this mode was positively associated with higher nervous and worrier feelings, but no other emotional regulation issues and risk-taking behaviors. Importantly, subtype 3 did not indicate any positive associations in DiaWAS and MedWAS - an observation compatible with a lower prevalence of health issues.

## Subtype 4: a more female-biased morningness pattern

Subtype 4 uncovered another morningness pattern, as evidenced by significantly higher brain scores in the morningness group (t(27,028) = 6.74, $P < 0.001$). This subtype also showed that females exhibited significantly higher brain scores than males (two-sample t-test for females > males: t(27,028) = 16.72, $P < 0.001$). In addition, this subtype was linked to female-specific health traits in DiaWAS, such as disorders of menstruation, further suggesting a predominantly female expression pattern. The variations in the regional GMV revealed positive loadings in subcortical areas including the putamen, ventral striatum, caudate, hippocampus, thalamus, amygdala, as well as whole cerebellum, suggesting that greater GMV in these regions contributed to stronger expression of this early bird subtype. Conversely, negative loadings in the temporal cortex implied that lower GMV in these regions was also associated with this early bird subtype (Fig. 4a, Supplementary Fig. 4a). The integrity measures of major white matter tracts uniformly showed negative loadings, indicating that reduced white matter integrity in these tracts was associated with stronger expression of this morningness subtype (Fig. 4b). At the functional connectivity level, connections between the visual and somatosensory cortex had positive loadings, suggesting that stronger coupling between these neural circuits supported this early bird subtype. In contrast, the couplings between the basal ganglia/cerebellum and somatosensory cortex had negative loadings, implying that weaker connectivity between these regions was linked to this morningness subtype (Fig. 4c, Supplementary Fig. 4b).

With respect to PheWAS (Figs. 4d–f, Supplementary Table 2), our results showed that subtype 4 was associated with lower blood testosterone levels and no hair balding patterns, as well as higher level of sex hormone-binding globulin (SHBG) and depression symptoms. In DiaWAS, this mode was associated with menstruation disorders, and in MedWAS, it showed links to intake of antidepressants, analgesics, antipyretics, and calcium supplements. Overall, subtype 4 delineated a female-dominant pattern, characterized by morningness traits and associations with lower testosterone levels, menstruation disorders, and depression phenotypes.

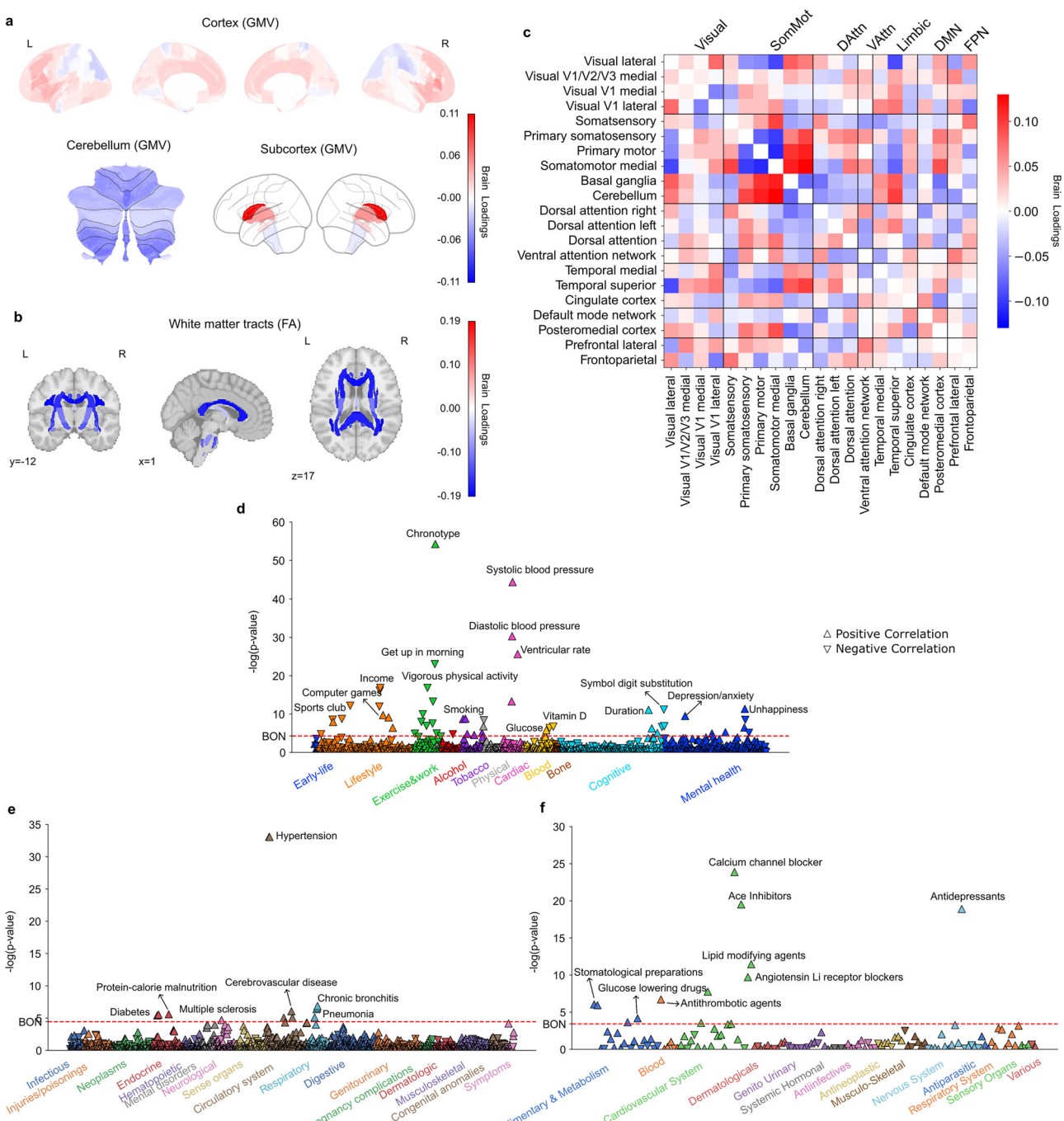

**Fig. 2 | Subtype 2 is a night owl pattern associated with decreased white matter integrity, smoking, cardiovascular risks, lower physical activity, as well as depressive symptoms and antidepressant drugs. a** Brain loadings of GMV in cortex, subcortex, and cerebellum, highlighting the basal ganglia, thalamus, and cerebellum. **b** Brain loadings of FA of white matter microstructures, highlighting lower integrity for all white matter tracts. **c** Brain loadings of functional connectivity between 21 meaningful ICA components. Abbreviations of networks see Fig. 1c. **d–f** The phenome-wide association study with behaviors, diagnoses and medications. Pearson's correlation results for each phenotype are presented on a logarithmic scale of associated *P*-values. The horizontal line denotes the significance thresholds following Bonferroni correction (*P* < 0.05). Full details are provided in Supplementary Table 2. Source data are provided as a Source Data file.

## Subtype 5: a more male-biased eveningness pattern

Our last brain-extracted subtype shed light on further previously overlooked nuances in chronotype, showing higher brain scores in the eveningness group (t(27,028) = -2.79, *P* = 0.005), indicating that greater expression in this subtype corresponded to eveningness. Moreover, within this subtype, females had significantly lower brain scores than males (t(27,028) = -30.22, *P* < 0.001). Its positive associations with male-related phenotypes, such as hair bald patterns and prostate diseases, further underscore this subtype's male-biased profile. At GMV level, this subtype showed positive loadings in basal ganglia (e.g., caudate, ventral striatum and putamen), thalamus, hippocampus, and primary visual areas (e.g., intracalcarine cortex and supracalcarine cortex), suggesting that greater GMV in these regions contributed to stronger expression of this night owl subtype. On the contrary, negative loadings were observed in the left inferior frontal gyrus, right precentral gyrus, and left superior parietal lobule indicated that lower GMV in these regions also supported this eveningness pattern (Fig. 5a, Supplementary Fig. 5a). Regarding white matter

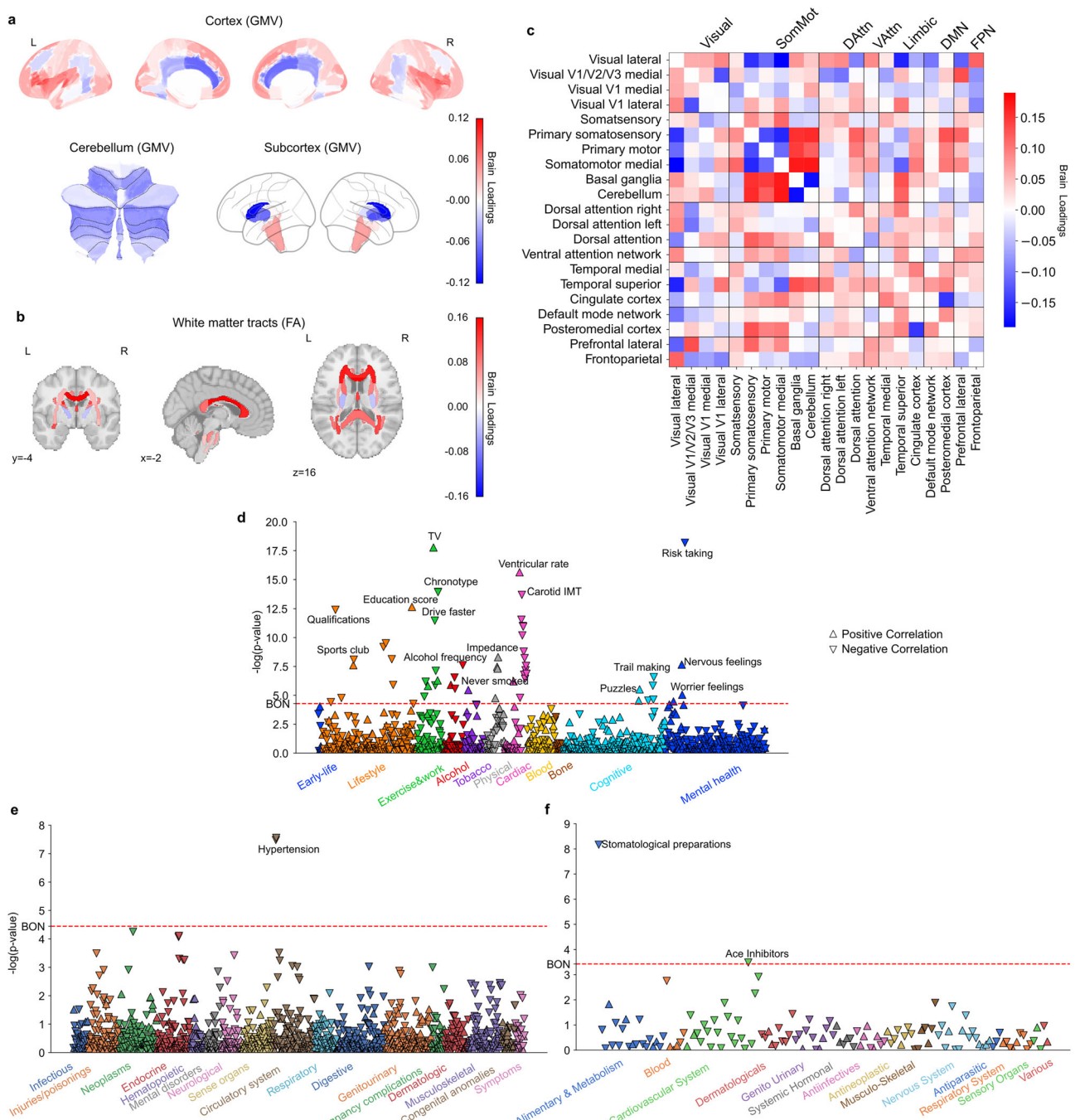

**Fig. 3 | Subtype 3 is an early bird pattern associated with education, non-smokers, rare alcohol intake, low risk taking and fewer emotional issues.** **a** Brain loadings of GMV in cortex, subcortex, and cerebellum, highlighting the basal ganglia, limbic cortex, frontal cortex. **b** Brain loadings of FA of white matter microstructures. **c** Brain loadings of functional connectivity between 21 meaningful ICA components. Abbreviations of networks see Fig. 1c. **d–f** The phenome-wide association study with behaviors, diagnoses and medications. Pearson's correlation results for each phenotype are presented on a logarithmic scale of associated *P*-values. The horizontal line denotes the significance thresholds following Bonferroni correction (*P* < 0.05). Abbreviations: IMT: intima-medial thickness. Full details are provided in Supplementary Table 2. Source data are provided as a Source Data file.

integrity, the cingulum hippocampus tracts, corticospinal tracts, and cerebral peduncle tracts showed significant positive loadings, suggesting that increased integrity in these tracts was associated with more expression of this subtype (Fig. 5b, Supplementary Fig. 5b). Conversely, lower integrity in fornix was associated with this subtype's expression. At the functional connectivity level, couplings between ventral attention network and cingulate cortex, as well as between the prefrontal lateral cortex and posteromedial cortex exhibited positive loadings. In contrast, connections between the dorsal attention

network and medial temporal cortex, as well as within-network links in the default mode network, tended to show negative loadings (Fig. 5c, Supplementary Fig. 5c).

In PheWAS (Figs. 5d–f, Supplementary Table 2), subtype 5 was positively associated with hair balding patterns, alcohol and cigarette consumptions, blood pressure, testosterone levels, risk-taking, and cannabis use, while it was negatively associated with SHBG levels and depression symptoms. In DiaWAS, consistent with a male-biased source of population variation, this mode was positively linked to

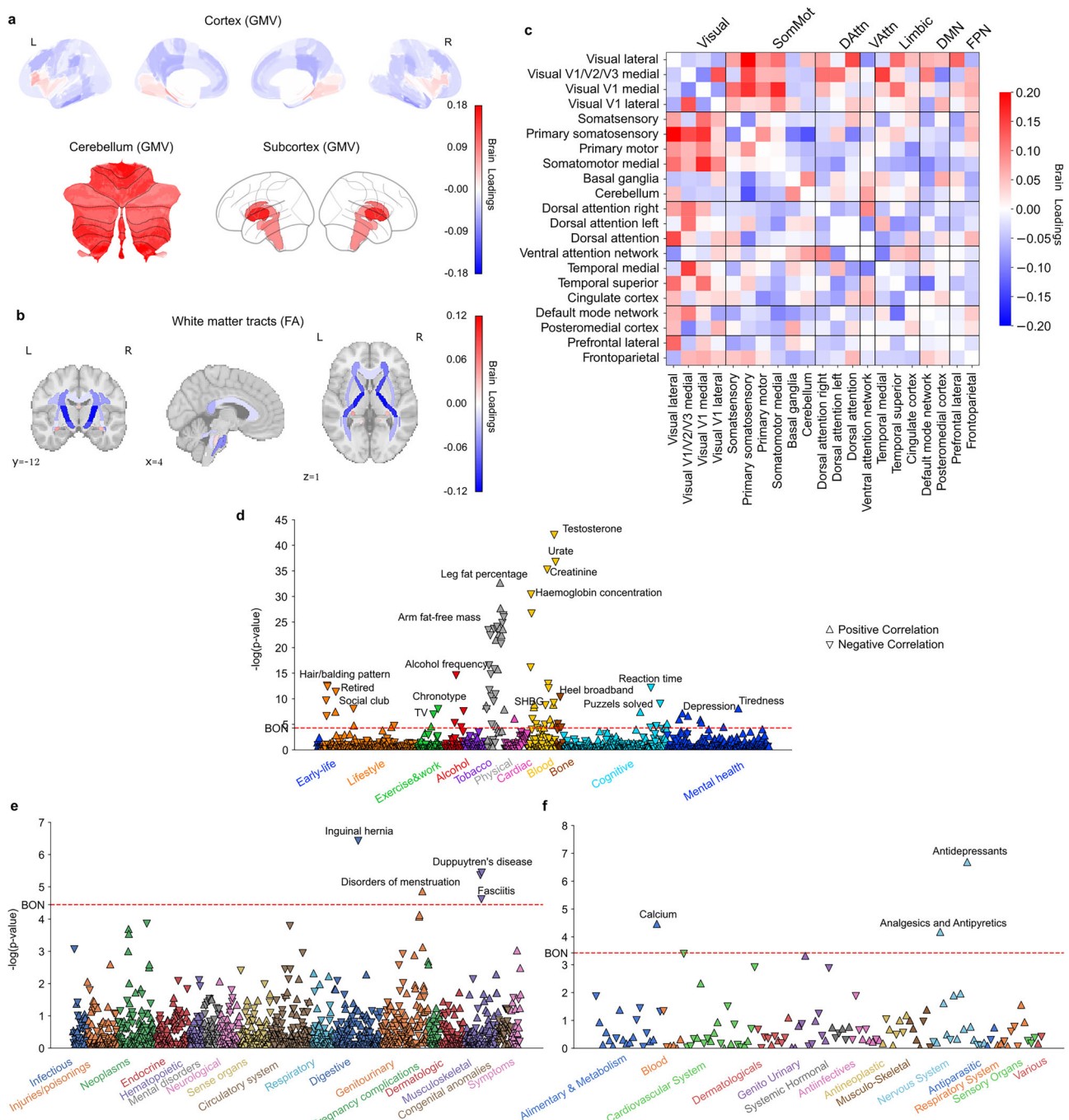

**Fig. 4 | Subtype 4 is a female-dominant early bird pattern linked to depressive symptoms and antidepressant drug prescriptions. a** Brain loadings of GMV in cortex, subcortex, and cerebellum, highlighting the basal ganglia, thalamus, temporal cortex, and cerebellum. **b** Brain loadings of FA of white matter microstructures. **c** Brain loadings of functional connectivity between 21 meaningful ICA components. Abbreviations of networks see Fig. 1c. **d**–**f** The phenome-wide association study with behaviors, diagnoses and medications. Pearson's correlation results for each phenotype are presented on a logarithmic scale of associated *P*-values. The horizontal line denotes the significance thresholds following Bonferroni correction (*P* < 0.05). Abbreviations: SHBG: Sex hormone-binding globulin. Full details are provided in Supplementary Table 2. Source data are provided as a Source Data file.

prostate cancer, prostate hyperplasia. In addition, it was also associated with hypertension, and inguinal hernia. In MedWAS, there were many cardiovascular system drugs that were positively associated with this subtype, including lipid modifying agents, selective calcium blockers, ace inhibitors, and cardiac glycosides. Moreover, subtype 5 was associated with medication drugs used for benign prostatic hypertrophy (further reinforcing a male subtype), antithrombotic agents, stomatological preparations, and antiglaucoma preparations and miotics.

**External validation to a different, younger cohort in the US**

Chronotype is known to play a role across the lifespan[20–22]. To validate our middle-to-old age group derived subtypes in a youth group, we applied the *already trained* PLS model from the UK cohort to a largest-of-its-kind cohort of youth - the ABCD Study. To that end, we distilled the GMV parameters pretrained on the UK Biobank and applied them to the ABCD Study GMV data to obtain the brain scores of the adolescents (see Methods for details). In other words, by carrying over the trained pattern learning model from one source cohort (UK Biobank)

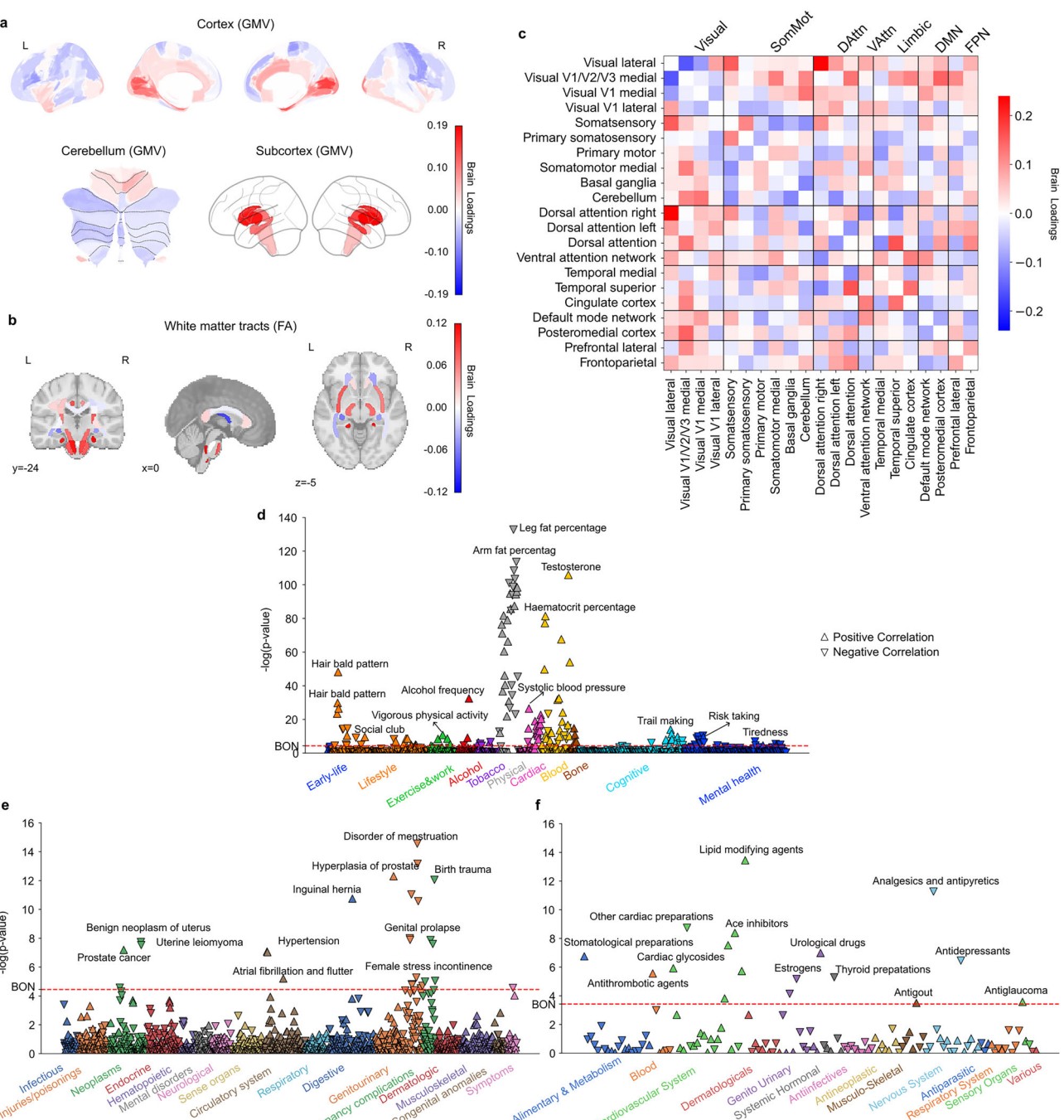

**Fig. 5 | Subtype 5 is a male-dominate night owl pattern associated with cardiovascular risks. a** Brain loadings of GMV in cortex, subcortex, and cerebellum, highlighting the basal ganglia, thalamus, and hippocampus. **b** Brain loadings of FA of white matter microstructures. **c** Brain loadings of functional connectivity between 21 meaningful ICA components. Abbreviations of networks see Fig. 1c. **d–f** The phenome-wide association study with behaviors, diagnoses and medications. Pearson's correlation results for each phenotype are presented on a logarithmic scale of associated P-values. The horizontal line denotes the significance thresholds following Bonferroni correction (P < 0.05). Full details are provided in Supplementary Table 2. Source data are provided as a Source Data file.

to an independent target cohort (ABCD Study)[39–41], we were able to derive the latent representation expressions (brain scores) for each of the participants in the ABCD Study cohort – the basis for all subsequent analyses on this unseen validation cohort. Notably, validation based solely on sMRI in the ABCD Study cohort would reduce the power of the holistic model derived from multi-modal data. Despite this likely loss in power, we validated the majority of our UKB-derived chronotypes in the ABCD Study.

Following the same steps as the UK Biobank analysis, we analogously conducted the PheWAS using these brain scores on 5859 total

ABCD Study phenotypes - a completely independent but partly conceptually related phenome. Overall, we identified five distinct PheWAS patterns for the ABCD Study cohort (Fig. 6, Supplementary Table 3). Mirroring our UK cohort findings, the first subtype was positively associated with cognitive performance tests, such as the N-back task and the Wechsler Intelligence Scale for Children (WISC-V, fifth edition). This mode was also positively associated with age in months, suggesting that children in this mode were relatively older. The second subtype showed numerous hits related to the Kiddie-Adolescent Schedule for Affective Disorders and Schizophrenia (KSADS) in

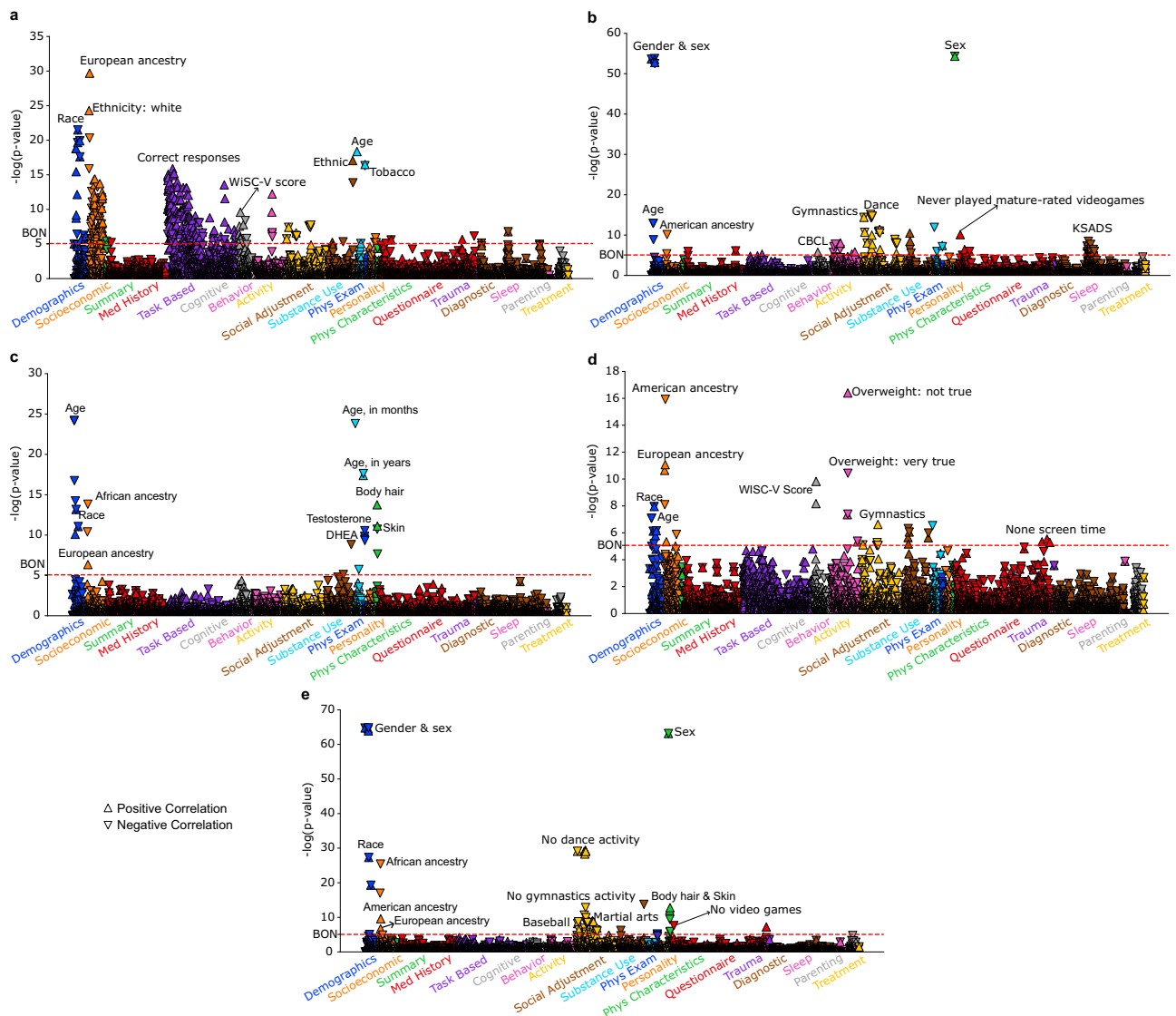

**Fig. 6 | Five modes derived from the ABCD Study cohort reveal distinct phenotypic associations.** Latent representations for the ABCD Study cohort were obtained by applying the UK Biobank-trained model to the sMRI data derived from the participants in the ABCD Study cohort. These representations were used to conduct the PheWAS across 5,859 phenotypes spanning 19 categories. Pearson's correlation results are presented on a logarithmic scale of associated *P*-values. The horizontal line denotes the significance thresholds following Bonferroni correction ($P < 0.05$). Full details are provided in Supplementary Table 3. **a** Mode 1: associated with better cognitive abilities. **b** Mode 2: female-dominated pattern with diagnostic hits indicating good mental health. **c** Mode 3: linked to younger age and lower levels of testosterone and DHEA. **d** Mode 4: associated with female-related activities and no screen time. **e** Mode 5: linked to male-related activities and playing mature-rated video games. Source data are provided as a Source Data file.

diagnostic categories. Interestingly, these subtype-phenotype hits indicated that this mode is associated with more favorable diagnostic outcomes, in contrast to the depression-related symptoms observed in UK adults. Moreover, this mode was significantly associated with sex (female) and more female-prevalent activities, such as gymnastics and ballet. For the third subtype, the predominant association was with age, which was negatively associated, indicating that this mode is associated with a younger age. Moreover, this mode exhibited lower levels of testosterone and Dehydroepiandrosterone (DHEA). Subtype 4 was associated with more female-prevalent activity (gymnastics), higher WISC-V scores, and no screen time. Subtype 5 was significantly correlated with more male-prevalent activities, such as baseball, martial arts and playing mature-rated video games, and negatively associated with more female-prevalent activities, including ballet and gymnastics. In addition, with the exception of subtype 2, all other subtypes in the children cohort were associated with European ancestry, while subtype 5 was also associated with American ancestry.

Overall, using the same chronotype-associated sMRI brain patterns, we identified similar activity associations in the young ABCD Study cohort, particularly in cognitive tests and sex-associated activities. Notably, the ABCD Study cohort from the US exhibited fewer mental health associations compared to our middle-to-old age cohort from the UK.

**Shifts in chronotype subtypes across the lifespan**
We finally examined the age distribution of participant-specific expressions of the five subtypes for both cohorts. Specifically, we categorized the UK (Biobank) participants into five age brackets: 40−50, 50−55, 55−60, 60−65, 65−70, based on their reported ages. Similarly, we grouped the US (ABCD Study) participants into four groups: 100−110 months, 110−120 months, 120−130 months, 130−140 months. We then obtained the mean brain scores of each age group for the 5 subtypes, which higher scores indicated a stronger inclination toward a given subtype. Measuring the presence of

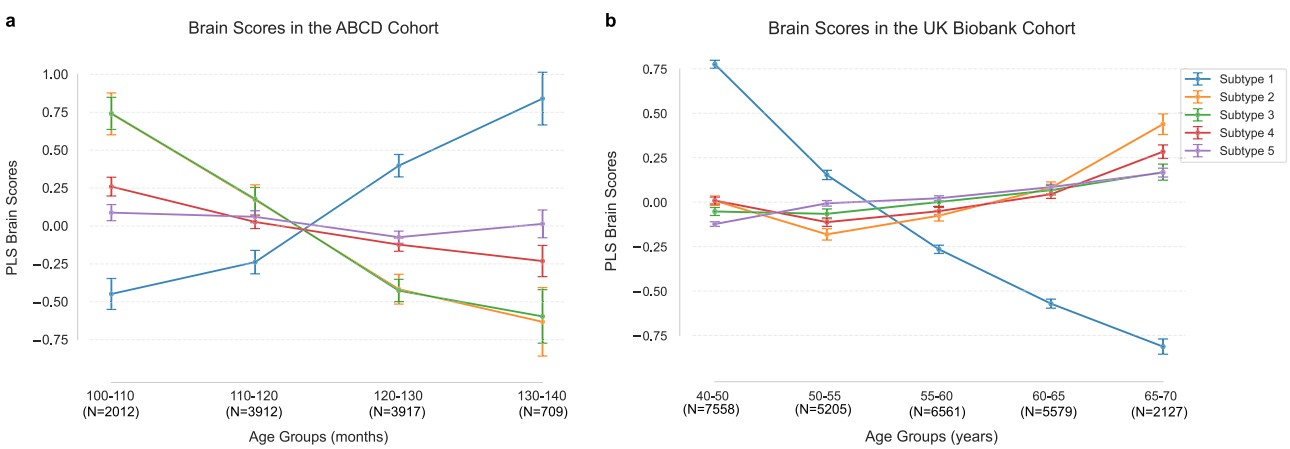

**Fig. 7 | The age distributions in the UK Biobank and ABCD Study cohorts show reverse patterns across subtype expressions.** Latent representations (brain scores) were grouped by age to visualize age distributions for 5 subtypes in the UK Biobank and ABCD Study datasets. **a** Age distributions for the ABCD Study cohort.

**b** Age distributions for the UK Biobank cohort. Each dot represents the mean brain scores within a specific age group, with error bars indicating standard deviations. Source data are provided as a Source Data file.

5 subtypes definitions in each adult vs. teenager, the two cohorts showed reverse age distributions across the five distinct modes (Fig. 7).

For the first night owl mode, the UK Biobank exhibited higher mean brain scores in the relatively younger group (40–50 years old, brain scores: 0.775 +/− 0.022). Instead, the ABCD Study cohort showed higher mean brain expressions in the relatively older group (130–140 months, brain scores: 0.840 +/− 0.174). For the other modes in the UK Biobank cohort, higher mean brain expressions were in the older age groups (65–70 years old). In contrast, for the ABCD Study cohort, the other subtypes showed higher mean individual brain expression in relatively younger groups (100–110 months old), particularly for the morningness mode (subtype 3) and depression-smoking subtype (subtype 2). Overall, cross-cohort comparisons demonstrated reversed age-dependent expression patterns, with night owl and early riser subtypes showing opposing distributions between our adult participants in the UK and our youth participants in the US.

## Discussion

Chronotype is a multipronged construct that is increasingly realized to be closely tied into behaviors and health outcomes[5–8]. Previous studies implicitly assumed that people with a given chronotype form a compact group of coherent phenotypic characteristics. This contention is potentially overlooking the rich variability in brain and behavioral patterns that can exist among individuals with a similar chronotype. An adequately nuanced conceptualization and reconceptualization towards a more fine-grained chronotype spectrum definition can help optimize performance schedules of our workforce, mitigate potential health risks, and elevate well-being across different age groups and society as a whole. To fill this gap and reconcile previous inconsistency in chronotype research, we developed a dedicated analytical framework grounded in a purposeful supervised pattern-learning analysis for subtyping at the population level.

By co-analyzing three complementary ways to capture brain macroscopical features and deep phenotypic fingerprints from 27,030 UK Biobank participants, we identified five unique subtypes of chronotype-brain covariation, each with specific behavioral and clinical determinants. The first mode was a night owl subtype primarily linked to younger age and emotional difficulties, including irritability/mania, tiredness, and self-harm. The second mode tracked variation in a different subtype of night owls with professionally diagnosed depression and cardiovascular risks or diseases. The third mode outlined an early bird subtype connected with relatively better mental health (limited to nervous and worrier feelings) and overall healthier cardiovascular status. The fourth mode was a more female-expressed

early bird subtype correlated with depressive symptoms and female-special associations (e.g., lower testosterone level, higher SHBG level, menstruation disorders). Lastly, the fifth mode was a more male-expressed night owl subtype linked to elevated cardiovascular risks and male-special associations (e.g., higher testosterone level, lower SHBG level, prostate diseases). In an independent model validation, we applied the UK cohort derived predictive model to 10,550 participants from the ABCD Study cohort, which comprises data collected from younger adolescents in a different country—the USA. In fact, the results revealed opposite age patterns across the five subtypes compared to the original UK adult cohort. For example, the typical night owl subtype 1 which more prevalent in younger adults in the UK cohort, was more frequently expressed in older adolescents in the ABCD Study sample, while the early bird subtype 3 showed the reverse trend. This inversion of subtype occurrences along age was accompanied by the better cognitive performance in the night owl subtype 1. Furthermore, consistent sex-associated behavioral patterns observed in subtypes 4 and 5 also persisted across different age groups. Our collective findings highlight the heterogeneity across chronotype by deriving and characterizing biologically grounded subtypes of chronotype variation in the wider, diverse population, underscoring the need for more precise investigations to inform future chronotherapeutic and other lifestyle-improvement approaches.

Numerous previous research reports associated evening chronotype to emotional regulation deficits[27,30,42–45]. Our subtype-resolving analysis showed that night owl subtype 1, in particular, was associated with irritability/mania, and motivational deficits (e.g., unenthusiasm and fed-up feelings). Interestingly, several functional and structural brain regions that were associated with subtype 1 are implicated in emotional processing circuits in and around the limbic system, including the amygdala, orbitofrontal cortex (OFC), and anterior cingulate gyrus[46,47]. Furthermore, aligning with previous observations[48], individuals of this subtype were more likely to engage in substance dependence behaviors, such as smoking, alcohol consumption, and cannabis use. In contrast, our approach more precisely localized one early bird pattern (subtype 3) to fewer emotional disturbances and lower levels of alcohol and cigarette consumption, as well as distinct intrinsic functional coupling manifestations in limbic networks, and basal ganglia. These findings further underscore the divergent behavioral and neural profiles of chronotypes. Notably, these neurobiological underpinnings were evident in brain regions linked to reward processing, which are known to play a central role in addictive behaviors[49–52]. This constellation suggests a potential link between chronotype-related brain variations and susceptibility to addiction.

Moreover, our night owl pattern (subtype 1) was linked to higher GMV of frontal regions, greater white matter integrity, as well as stronger connectivity among attention and frontal networks. These findings align with previous studies linking eveningness to higher cognitive abilities[53–55]. As our PheWAS results showed, some cognitive phenotypes also support these cognitive-performance-related findings, such as indexed by puzzle-solving performance and symbol digital substitution performance.

In addition, across extracted modes, Vitamin D, a key blood marker whose level is known to be mediated by light exposure, was negatively associated with the eveningness subtypes but not the morningness subtypes, further solidifying the discriminatory power of our subgrouping approach. Collectively, consistent with previous findings, our analyses identified typical eveningness and morningness patterns, highlighting the key differences in emotional regulation, substance dependence, cognitive abilities and neurobiological basis between those two chronotypes from canonical research. Importantly, we move beyond an overly strict binary framework by sorting previously reported phenotype associations into coherent, discernable chronotype subtypes that can be reliably identified in one of the highest quality population data resources available to date.

Previous small-sample, subgroup-naive studies on chronotypes have shown that evening types are broadly linked to exacerbated vulnerability for physical and mental health problems, such as depression, anxiety, and cardiovascular diseases[48,56–59]. However, the inter-relations between chronotype and health risks demonstrate substantial heterogeneity across studies, with some earlier studies linking night owls to risk of depression while other studies fail to support this association[38]. Beyond the typical night owl subtype 1 (associated with general emotional deficits), our more nuanced analytical approach identified a distinct night owl pattern (subtype 2) linked to professional diagnosed depression. Individuals with higher scores in subtype 2 showed proneness to experiences of depression and anxiety, smoking, antidepressant use, and cardiovascular comorbidities (including hypertension, diabetes, chronic bronchitis, and cerebrovascular diseases), with consistency across behavioral, diagnostic, and medication-based metrics. Previous studies have shown that individuals with depression and anxiety were more likely to smoke, and nominated smoking as a key mediating pathway that may link depression and anxiety to the aforementioned health issues, such as chronic bronchitis, heart diseases, and lung cancers[60,61]. Moreover, both depression and smoking have been consistently linked to cardiovascular risks before[62–64]. At the brain level, reduced microstructural integrity in major white matter fiber tracts, spatially spread out across the brain, emerged as the dominant brain phenotype captured by subtype 2 and has been previously tied to smoking and hypertension[65,66]. Collectively, these findings position subtype 2 as a unique chronotype subtype that specifically captures co-occurring depression, smoking, and cardiovascular risks. This further underscores the heterogeneity in chronotype-health associations in different segments of the wider human populations. For these reasons, unresolved inconsistencies in reports and conclusions from the existing chronotype literature regarding the link between depression and night owls may be reconciled through refinement of chronotype subtype definitions, such as for application in future randomized clinical trials informed by biologically valid subtype definitions.

Further, our subtype-resolving framework has also differentiated important sex differences in chronotype, which may arise from a combination of biological, hormonal, and social factors. For example, hormonal fluctuations across the menstrual cycle in females are known to contribute to variations in behaviors influenced by sleep-wake rhythm[67,68]. Additionally, females are known to have earlier chronotypes on average, as well as earlier entrained circadian rhythms of core body temperature and melatonin secretion[23,69–71]. Our analytical approach indeed disclosed two subtypes that were especially sex-

associated: a more female-expressed morningness pattern (subtype 4) and a more male-expressed eveningness pattern (subtype 5). The sex-biased hormonal profiles between these two subtypes further support their associations with sex: female pattern (subtype 4) was characterized by lower testosterone and higher SHBG levels, whereas male pattern (subtype 5) exhibited higher testosterone and reduced SHBG. This sex difference in hormonal fluctuation aligned with the known inverse relationship between SHBG and testosterone, as SHBG plays a pivotal role in modulating the bioavailability of testosterone, with higher SHBG levels leading to reduced free testosterone[72]. It came as no surprise that the bottom-up pattern-learning approach here found chronotype subtypes with specific ties to variation in testosterone levels. In fact, testosterone has been shown to be closely related to sleep phenotypes in males and its accumulation occurs during sleep, regardless of whether subjects slept during the day or night[23]. Interestingly, testosterone has also been linked to sleep patterns in women; those with higher testosterone levels tend to sleep more continuously after falling asleep[73].

Moreover, testosterone was previously reported to be positively correlated with chronotype in males but negatively correlated with chronotype in females[74,75], which confirmed and detailed our present findings with phenome-wide subtype profiling. Notably, previous studies on sex differences have shown that females are twice as likely to encounter depressed or anxious states than males and suggested that testosterone plays a potential role in the etiology of depression and anxiety[76,77]. In line with these findings, it was precisely our identified female subtype (mode 4) that exhibited associations with depression phenotypes (PheWAS) and, consistently, the use of antidepressant drugs (MedWAS). The positive association with menstruation disorders (DiaWAS) further suggests heightened psychiatric risks for women in this subtype, given the established links between menstrual dysfunction and broader mental health symptoms[78]. In contrast, the male biased subtype 5 tied to fewer depression-related symptoms but a greater propensity for risk-taking, hair baldness, risks of cardiovascular complications, and prostate cancer - all of which are known to be influenced by higher testosterone levels[79–81].

Furthermore, sex hormones are known to play a crucial role in human physiology and mental health, with broad influences on sex differences in the brain, especially through brain circuits such as the hippocampus, prefrontal cortex, and the dopaminergic neurotransmitter systems (e.g., caudate nucleus, nucleus accumbens)[82]. Consistent with these findings, the sex-associated brain patterns in our data also highlighted the key involvement of the brain regions, including the thalamus, caudate, putamen, ventral striatum, and hippocampus. Importantly, these two distinct sex-associated subtypes of chronotype behavior, identified in the UK Biobank cohort, were replicated in the younger ABCD Study cohort. Subtype 5 was strongly associated with males and traditionally more male-prevalent activities (e.g., baseball, martial arts and play mature-rated video games), while subtype 4 was linked to more female-prevalent activities (e.g., gymnastics). Overall, our approach precisely identified a subset of chronotype categories that was closely related to sex differences, which we contextualize as a function of hormone status differences at the population scale. These differences were also evident in brain patterns and could be detected even at a much younger age, in teenagers, and in a cohort from a different continent (US) than the one in which the chronotype subtypes were originally identified (UK).

Chronotype behavior of a particular individual is known to change across the lifespan: being an earlier riser in early childhood, delaying to become more of a night owl during adolescence, and gradually shifting back to earlier timing with aging[20–22]. Consistent with these earlier observations of chronotype patterns being in a characteristic flux throughout life, our findings in middle-to-old age (UK Biobank) revealed that the eveningness pattern (subtype 1) was predominant in relatively younger adults (40–50 years) in the UK Biobank cohort,

while this same mode was more commonly apparent in relatively older adolescents (130–140 months of age) in the ABCD Study youth cohort. Notably, in the ABCD Study cohort, we found this eveningness subtype was also associated with higher cognitive abilities (e.g., n-back task performance; higher WISC-V scores), in agreement with our findings on conceptually related phenotypes in the UK Biobank cohort (e.g., puzzle-solving, symbol digital substitution). Interestingly, subtype 2, which was identified as a depression and cardiovascular risk-associated pattern in the UK Biobank dataset, emerged as another female-associated pattern in the US youth sample (ABCD Study dataset). These divergences may stem from neuroimaging data differences as our ABCD Study analysis built on structural MRI (T1), while diffusion MRI measurements dominated the UK Biobank findings of this mode. Another possible explanation is the relatively lower prevalence of any mood disorders (including depression and bipolar) among children in the ABCD Study cohort (3.11%, $N = 11,874$)[83], compared to the higher prevalence observed in the UK biobank cohort (28.2%, $N = 123,000$)[84]. Moreover, as discussed earlier, the sex-associated chronotype subtypes (subtype 4 and subtype 5) were replicated in the youth ABCD Study cohort, suggesting that potential sex differences in chronotype may emerge at an early stage of life[20]. Collectively, by extending our analysis from UK adults to US youth, our findings confirmed age-related shifts in chronotype and revealed shared sex-related characteristics in the brain-chronotype relationship across different age groups.

Certain limitations of the current study should be noted. While questionnaire-based measurements of chronotype have been shown to be reliable and robust[85–88], subjective factors may introduce potential confounding that could contaminate our findings. Future investigations could benefit from more directly incorporating objective measures, such as actigraphy data from wearable devices, mid-sleep time or blood melatonin fluctuations, to capture a more nuanced and dynamic representation of chronotype[85,86,89]. Additionally, by jointly analyzing findings from two unique, largest-of-its-kind population cohorts, the UK Biobank and ABCD Study datasets, the use of multiple data modalities provided a more complete picture on the brain-behavior basis of chronotype in humans. To further enhance consistency and comparability across studies involving different age groups, future research could enhance our present collection of findings by integrating additional modalities across all cohorts, as well as further cohort covering more age brackets and broader population diversity by including sample from countries beyond US, UK and the Western world[90,91]. In addition, our cross-sectional approach captured a snapshot rather than the temporal dynamics of these subtypes. Future longitudinal studies will be valuable for revealing transitions and dynamic changes both between and within subtypes, which would further enable early detection and preventive interventions before maladaptive behaviors or pathological processes become established.

Notably, our data-driven approach aims not to create a discrete diagnostic clinical tool. Instead, we identified latent neurobiological dimensions of chronotype that exist in the population, which could pave the way for personalized interventions by identifying specific risk profiles and biological pathways for future validation in dedicated interventional or animal studies. Chronotypes themselves are not disorders that require treatments, but some disorders may relate to certain components of chronotype, their practical interventions may also benefit from our framework. For instance, the distinct depression-associated profiles we identified suggest different therapeutic strategies. A depressed patient with the mood-associated eveningness subtype 2 profile might respond best to an integrated approach combining circadian realignment (e.g., bright light therapy[92]) with physical activity protocols. Conversely, a depressed patient with morningness subtype 4 profile, which was linked to female-related factors, reduced social engagement, and sexual assault, might benefit more from interventions focused on enhancing social support and

connectivity. Furthermore, to avoid potential confounds arising from circadian misalignment (social jet lag), our study excluded participants with history of shift work. Future studies specifically designed to capture circadian misalignment will be a valuable next step for refining these profiles in this specific population and for advancing future clinical applications.

In conclusion, by combining multiomic datasets from both adolescent and middle-to-old-life age groups with a purposefully devised analytical framework, we revealed substantial heterogeneity in brain-chronotype-behavior patterns, as well as their complex interactions with age and sex. As a consequence of our present findings and conclusions, it may become increasingly evident that our internal sleep phasing system, of which the circadian clock is likely a major component, ties into many more facets of daily life than previously assumed.

## Methods

### UK Biobank population data resource

Our selected population cohort constitutes a prospective epidemiological cohort providing comprehensive behavioral and demographic assessments, medical and cognitive measures, and biological samples from over 500,000 participants across the United Kingdom (https://www.ukbiobank.ac.uk/; application number for our study: 25163). The present study was based on the UK Biobank data released in February 2020, involving over 40,000 participants, with brain-imaging assessments. Participants were identified (i.e. those aged 40–69 years and who lived within ~25 miles of an assessment center) and invited through the National Health Service (NHS). Participants were not given financial incentives, and all of them provided informed consent. Ethical approval has been granted to the UK Biobank project, ensuring the study's adherence to high ethical standards (https://www.ukbiobank.ac.uk/learn-more-about-uk-biobank/governance/ethics-advisory-committee). Additional information on the consent procedure is available here (https://biobank.ctsu.ox.ac.uk/crystal/field.cgi?id=200).

The main target phenotype in this study was 'chronotype' (data-field 1180). It was assessed in the UK Biobank cohort through the following self-report question answers: "Do you consider yourself to be?" Participants chose from six options: "definitely a morning person", "more a morning than an evening person", "more an evening than a morning person", "definitely an evening person", "do not know", and "prefer not to answer". To enhance interpretability and align with our previous study[9], we defined a binary variable for chronotype within the same data field. Participants who selected "definitely a morning person" or "more morning person than an evening person" were encoded as "early birds" (henceforth coded as –1), those who chose "definitely an evening person" or "more an evening than a morning person" were labeled as "night owl" (henceforth coded as 1). Moreover, we conducted sensitivity analyses by comparing models with binary and the raw 4-level encoding of the chronotype variable. The derived scores and loadings were significantly correlated between two models (all Pearson's $r > 0.9$, $P < 0.001$, see Supplementary Fig. 7), indicating the core multivariate patterns derived in our main analysis are robust and not meaningfully altered by our binary target phenotype. The UK Biobank data is structured into four specific 'instances' that represent different time points for data collection. These instances include the initial assessment visit (2006-2010, instance 0), the follow-up assessment visit (2012-2013, instance 1), the imaging visit (2014 +, instance 2), and the first repeat imaging visit (2019 +, instance 3). To enhance comparability and reproducibility, subjects displaying disparities in chronotype assessment (coded as –1 or 1) between the initial visit (data field 1180-0.0) and the first brain-imaging visit (data field 1180-2.0) were excluded. Furthermore, to avoid contamination by effects due to shift work, individuals with a history of shift work (defined as jobs involving shift work with data-fields 826-0.0 and 826-2.0, and jobs involving night shift work with data-fields 3426-0.0 and 3426-2.0) were excluded. Our study hence ultimately included 27,030 subjects with

neuroimaging-derived measures, covering i) gray matter volumes (GMV; T1-weighted MRI, sMRI), ii) white matter tract microstructures (diffusion MRI, dMRI), and iii) intrinsic functional connectivity (resting-state fMRI, rs-fMRI). In particular, 17,995 participants were identified as "early birds", and 9,035 participants were "night owls". Participants were recruited between the ages of 40 and 70 years (mean age with standard deviation: 55.37 ± 7.39 years), comprising 54.44% females (see Supplementary Table 1 for details).

The target brain measurements used in our study are drawn from three variable sets that were previously generated by a uniform imaging data processing and cleaning pipeline, which was designed and implemented by the Oxford Centre for Functional Magnetic Resonance Imaging of the Brain (FMRIB) at Oxford University, UK[93]. The specific protocol and processing pipeline are provided at: https://biobank.ctsu.ox.ac.uk/crystal/crystal/docs/brain_mri.pdf. The curated imaging-derived phenotypes (IDPs) were extracted following standard processing, encompassing (i) 139 volumes of cortical/subcortical grey matter (111 cerebral regions from the Harvard-Oxford atlas[94–97] and 28 cerebellum regions from the Diedrichsen atlas[98]), (ii) fractional anisotropy (FA) of 48 major white matter microstructures from the John-Hopkins University atlas[99–101], and (iii) 210 functional coupling link strengths (resting-state functional connectivity was based on robust 21 brain components of spatiotemporally coherent networks derived from an initial set of 25 whole-brain spatial independent component analyses (ICA), with four artifactual components removed[102,103]. Consistent with prior studies on the UK Biobank[104], nuisance variables that could potentially contaminate the interindividual brain variation of interest were regressed out: these variables were outside primary scientific interest and included body mass index, head size, head motion during task-related brain scans, head motion during resting-state fMRI scanning, head position (x, y, and z), position of the scanner table, as well as the data acquisition site[104,105]. Notably, building on a previous study that identified significant correlations in head motion across the three modalities, we regressed it out from all brain-imaging measures following their approach[104,105]. Additionally, to acknowledge potential confounding from sleep per se, which is commonly viewed to be distinct from chronotypes[106,107], we regressed out sleep duration (data-field: 1160-2.0), daytime napping (data-field 1190-2.0), sleeplessness/insomnia (data-field: 1200-2.0), snoring (data-field: 1210-2.0), and daytime dozing (data-field 1220-2.0) for all downstream analyses. We also conducted sensitivity analyses by comparing models with and without regressing out sleep phenotypes. The derived scores and loadings were significantly correlated between two models (all Pearson's $r > 0.9$, $P < 0.001$, see Supplementary Fig. 8), indicating negligible effects of sleep phenotypes on our main modeling. Sex and age were key factors or target variables of interest in our study; therefore, we did not regress them out. Moreover, our previous study has shown sex, age, and the time-of-day effect has negligible impacts on our UK Biobank sample and target MRI measurements[9].

The steps for sourcing behavioral phenotypic features were consistent with our previous study[108]. Initially, a raw collection of approximately 15,000 phenotypes was processed using the FMRIB UKBB Normalisation, Parsing And Cleaning Kit (FUNPACK version 2.5.0; https://zenodo.org/record/4762700#.YQrpui2caJ8). FUNPACK was employed to curate a refined set of phenotypes associated with the categories of interest and to perform data harmonization. The output from FUNPACK, comprising approximately 3,300 phenotypes, was subsequently input into the PHEnome Scan ANalysis Tool (PHE-SANT; https://github.com/MRCIEU/PHESANT) for further refinement, cleaning, and data categorization. This process yielded a final set of 977 phenotypes, including the chronotype phenotype.

Furthermore, to comprehensively understand the associations between chronotype and specific health outcomes, we employed an extensive range of ~1500 disease diagnoses from electronic health records[109]. Specifically, feature extraction for both ICD-9 and ICD-10

diagnoses was carried out using the FUNPACK (version 2.5.0; https://zenodo.org/record/4762700#.YQrpui2caJ8) to extract ICD codes. Once extracted, ICD-9 and ICD-10 codes were grouped into ~1,500 hierarchical phenotypes which combine related ICD-9 and ICD-10 codes into a single 'phecode', using previously established definitions[110]. In addition to combining related ICD codes, we used built-in exclusion criteria to reduce case contamination of control groups[111]. Moreover, phenotypes with constant value were removed in our 27,030 UK Biobank sample. The final set of 1396 clinical phecodes spanned 17 disease classes, ranging from congenital anomalies, and neoplasms, to mental disorders and infectious diseases.

The UK Biobank (UKB) medication data encompasses 6,745 distinct drug and supplement entries (data-field 20003). The information of medication was classified using the Anatomical Therapeutic Chemical (ATC) Classification System (www.who.int/tools/atc-ddd-toolkit/atc-classification), which categorizes active compounds based on their target organ systems along with their therapeutic, pharmacological, and chemical properties. This hierarchical system comprises five levels of specificity: organ system (Level 1), therapeutic purpose (Level 2), pharmacological mechanism (Level 3), chemical class (Level 4), and specific compound (Level 5). We particularly focused on Level 3 classifications to examine drug categories by their mechanistic actions. Specifically, all medication entries were processed through an automated mapping system that assigned appropriate ATC codes based solely on drug nomenclature regardless of formulation or strength[112]. Medications that could be classified into multiple ATC categories were assigned to the first available category in alphanumeric order. We excluded from analysis (1) medications without corresponding ATC codes and (2) combination drugs containing multiple active ingredients. The final analytic dataset comprised medication records mapped to 133 ATC Level 3 pharmacological categories across all 14 major Level 1 domains for our 27,030 UK Biobank participants.

## ABCD Study population data resource

To test whether derived patterns generalize to a younger cohort, we benefited from the largest and most comprehensive biomedical resource on child health and brain development, the Adolescent Brain Cognitive Development Study (ABCD Study®; https://abcdstudy.org/). Both brain imaging, and behavioral data in this study were obtained from 10,550 children aged 107−133 months (mean ± SD: 119.01 ± 7.51 months, 48.04% females) across 21 sites in the United States. All data were baseline measurements from the ABCD Study curated 4.0 release, providing comprehensive measures across child and parent domains, including self-reports of race and ethnicity, physical and mental health, neurocognitive performance, socio-demographic factors, cultural values, and environmental conditions[113]. All protocols for ABCD Study are approved by either a central or site-specific institutional review board committee[113]. Caregivers have provided written, informed consent and children provide verbal assent to all research protocols[114]. Additional information about the ABCD Study cohort can be found in the previous publication[115]. This dataset is administered by the National Institutes of Mental Health Data Archive and is freely available to all qualified researchers upon submission of an access request. All relevant instructions to obtain the data can be found at https://nda.nih.gov/abcd/request-access.

In addition, we included the structural MRI imaging (T1w) from the 10,550 participants from the ABCD Study cohort, based on analogous atlas definitions (111 cerebral regions from the Harvard-Oxford atlas and 28 cerebellum regions from the Diedrichsen atlas) that we used for the UK Biobank. The preprocessing performed using fMRI-Prep 21.0.2[116], which is based on Nipype 1.6.1[117]. A total of one T1-weighted (T1w) images were found within the input BIDS dataset. The T1-weighted (T1w) image was corrected for intensity non-uniformity (INU) with N4BiasFieldCorrection[118], distributed with ANTs 2.3.3[119], and used as T1w-reference throughout the workflow. The T1w-reference

was then skull-stripped with a Nipype implementation of the antsBrainExtraction.sh workflow (from ANTs), using OASIS30ANTs as target template. Brain tissue segmentation of cerebrospinal fluid (CSF), white-matter (WM) and gray-matter (GM) was performed on the brain-extracted T1w using fast (FSL 6.0.5.1:57b01774)[120]. Brain surfaces were reconstructed using recon-all (FreeSurfer 6.0.1)[121], and the brain mask estimated previously was refined with a custom variation of the method to reconcile ANTs-derived and FreeSurfer-derived segmentations of the cortical gray-matter of Mindboggle[122]. Volume-based spatial normalization to two standard spaces (MNI152NLin2009cAsym, MNI152NLin6Asym) was performed through nonlinear registration with antsRegistration (ANTs 2.3.3), using brain-extracted versions of both T1w reference and the T1w template. The following templates were selected for spatial normalization: ICBM 152 Nonlinear Asymmetrical template version 2009c[123], FSL's MNI ICBM 152 non-linear 6th Generation Asymmetric Average Brain Stereotaxic Registration Model[124]. To match the regions of interest we used in the UK Biobank cohort, we then warped the Harvard-Oxford atlas (include cortical and subcortical brain regions) and Diedrichsen atlas (cerebellum regions) from MNI space to T1w native space by implementing the transformation matrix derived from the preprocessing steps for each participant. By combining the warped atlas and derived gray matter segmentation, we obtained GMV measures for the 139 brain regions in the ABCD Study cohort, corresponding to the same regions used in the UK Biobank cohort. Similarly, we regressed out the total intracranial volume, scanning sites, and sleep phenotypes (elements name: from "sleepdisturb1_p" to "sleepdisturb26_p"), as these could account for interindividual variations in volume that might confound the primary focus of the study.

Furthermore, regarding the ABCD Study phenome, we leveraged the deep phenotypical variables of the 10,550 ABCD Study cohort participants, which spanned the breadth of the behavioral, clinical, cognitive, and socio-demographic information[125]. Phenotypes with data available for less than 80% of the total participants were excluded. To handle data columns with near-zero variance, phenotypes where the most common value appeared in >99% of cases compared to the second most common value were removed. Data processing workflows for discrete and continuous variables were handled separately for missing value imputation and further curation. Discrete data columns were one-hot encoded. For example, the variable "ethnicity" with categories White, Black, Hispanic, Asian, and Other was transformed into five binary encoded data columns: "White" represented as [1, 0, 0, 0, 0], "Black" as [0, 1, 0, 0, 0], "Hispanic" as [0, 0, 1, 0, 0], "Asian" as [0, 0, 0, 1, 0], and "Other" as [0, 0, 0, 0, 1]. Continuous data columns were standardized using robust z-scoring, consistent with previous research protocols[126]. Values exceeding four standard deviations from the mean (SD > 4) were adjusted to the largest value within four standard deviations of the mean (winsorization) to preserve data integrity and curtail the impact of outliers on model training. Finally, the complete repertoire of 5859 curated target phenotypes for the baseline time point, spanning 23 predefined categories, were used for PheWAS.

## Multivariate pattern learning protocol

The primary goal of the current study was to explore the potential existence of a series of distinct modes of covariation between brain and chronotype in the general population. Partial least squares (PLS) regression analysis, a multivariate pattern learning approach, is a natural choice for delineating relationships between high-dimensional sets of brain-derived variables and behavior phenotypes, by identifying latent structures that maximize the covariance between them[127]. In other words, the current study employed PLS as a supervised pattern-learning approach to distinguish the patterns associated with both brain and chronotype. The derived patterns were distinct from one another, as PLS extracts components sequentially, with each new component orthogonal to the previous ones while still capturing

information predictive of chronotype[128]. Furthermore, its ability to maximize the covariance structure of variable sets makes PLS particularly well-suited for analyzing high-dimensional, multimodal neuroimaging data with substantial known auto-correlation (e.g., covariates may occur within the same brain region across gray matter volume, white matter integrity, and functional connectivity). Specifically, for our UK Biobank study sample, the input variable set was constructed from the concatenated brain features, including 139 regional gray matter volume measures, 48 major white matter microstructure measures, and 210 functional coupling measures between 21 brain components, derived from 27,030 participants. The target variable set consisted of the dichotomized chronotype. As described in our previous study[9], the resulting dominant PLS components were evaluated for statistical robustness through a non-parametric permutation procedure. Specifically, in our study, the brain features were held constant while each row of the chronotype variable randomly shuffled, thus mixing group assignment between participants. This procedure generated a valid empirical null distribution of Pearson's correlation in the derived embedding space of the PLS model. It reflected the null hypothesis of a random association between brain features and chronotype. Pearson's correlation coefficients (rho) estimated between the perturbed low-rank projections were recorded in each iteration. P-values were derived based on the comparison of the original obtained rho and the 1000 rho estimates from the null PLS model, using the threshold $P < 0.001$. In other words, this criterion required that the association in the actual PLS model be greater than the model strength in all null PLS models where we knew there was no real association between brain and chronotype. This procedure guards against spurious noise findings. Overall, combined with empirical permutation schemes for statistical testing, our high-dimensional regression approach enabled the derivation of distinct and statistically robust patterns, which can serve as different modes or subtypes in the brain-chronotype relationship.

In order to quantify the relative importance of each brain feature for each of the derived significant PLS components, we implemented a bootstrapping resampling strategy for the PLS model to obtain the distributions of each brain feature loadings (PLS x-loadings). Concretely, in each bootstrap iteration, we created a perturbed dataset by randomly sampling the original dataset with replacement, maintaining the same sample size. The perturbed dataset was then used to fit a new PLS model. The order of the components was matched to the original PLS model's order using the Hungarian algorithm. Specifically, a correlation similarity matrix was computed between the bootstrapped brain loadings matrix (shape: loadings x components) and the original brain loadings matrix (shape: loadings x components). Based on this similarity matrix, the Hungarian algorithm identified the most similar pairs of components and reordered the bootstrapped PLS model components accordingly. Finally, we recorded the brain loadings of the bootstrapped PLS model using the same index as the significant components of the original PLS model. After 1,000 bootstrapping iterations, we obtained the distributions of brain loadings for all significant PLS components. Significant brain loadings were identified as those robustly different from zero, defined by a two-sided 5-95% confidence interval that excluded zero.

The latent brain representations (PLS x-scores) derived from the original data capture crucial individual differences in the embedded space, corresponding to each individual's relative position in the extracted latent brain components. In other words, these representations indicate how strongly each participant expresses or aligns with a given component, providing a quantitative measure of their alignment with the identified brain patterns. This feature is particularly advantageous for downstream analyses, as the latent representations can be integrated with additional behavioral and phenotypic data, such as those used in our phenome-wide association studies (PheWAS) and age distribution analyses. In summary, our approach is particularly

well-suited for studying factors that are inherently less variable (e.g., chronotype, defined here as eveningness and morningness) but are linked to multifaceted phenotypes, such as brain features, other behaviors, or health measures (See Supplementary Fig. 9 for overall workflow of the current study).

## Exploring the potential practical implications for each subtype

To explore the association between each PLS mode (subtype) and wide range of phenotypic characteristics in our UK Biobank population sample, we conducted three separate analyses: a behavioral phenome-wide association analysis (PheWAS), a diagnosis-wide association analysis (DiaWAS), and a medication-wide association analysis (Med-WAS). The phenotypes for each category were prepared as described in the previous section on population dataset resources. The latent representations (PLS x-scores) of significant components (subtypes) were derived from the PLS modeling. The strength of associations between phenotypes and subtypes was quantified using Pearson's correlation between PLS x-scores and the derived phenotypes. To adjust for multiple comparisons, we applied Bonferroni's correction for each analysis, with a significance threshold of $P < 0.05$. Specifically, the adjusted thresholds were $P < 0.05/977$ for PheWAS, $P < 0.05/1,396$ for DiaWAS, and $P < 0.05/133$ for MedWAS. For visualization purposes, each association was represented as the negative logarithm of the corresponding P-value, consistent with previous UK Biobank studies[103].

## Projection of ABCD Study Data into the UK Biobank latent space

To expand our analysis to a youth group, we applied the UK Biobank model to the ABCD Study cohort. The application of our adult-derived model to the child cohort allowed us to probe whether the observed brain-behavior relationships are stable across the lifespan or emerge specifically in the older adulthood. Since we could only work on the structural MRI for the ABCD Study cohort participants, we distilled the corresponding GMV parameters pretrained from the UK Biobank cohort and transformed the ABCD Study structural MRI data into x-scores. Specifically, the x-weights and x-loadings for the GMV features were derived from the UK Biobank PLS model, and the x-scores for the ABCD Study cohort were calculated as following formula[129]:

$$\mathbf{X}_{scores} = \mathbf{X}_{GMV} \cdot \mathbf{W} \cdot \left(\mathbf{P}^{T} \cdot \mathbf{W}\right)^{-1} \quad (1)$$

Where the $\mathbf{X}_{GMV}$ represents the GMV of the ABCD Study cohort participants, $\mathbf{W}$ denotes the weights from the trained model restricted to the GMV features, and $\mathbf{P}$ denotes the loadings from the trained model restricted to the GMV features. Overall, this approach allowed us to transform the ABCD Study data into the same latent space as the UK Biobank model while focusing exclusively on the GMV features. After applying the transformation, we obtained the latent representation expressions ($\mathbf{X}_{scores}$) for each participant in the ABCD Study cohort.

Following the same steps as the UK Biobank analysis, we conducted the PheWAS by integrating the brain latent representations and 5859 phenotypes from the 10,550 ABCD Study cohort participants. Pearson's correlations were calculated between the latent representations and phenotypes, with the Bonferroni correction at significance threshold $P < 0.05/5859$. The correlation results were presented using the negative logarithm P-value as UK Biobank results.

## Stratification of chronotype patterns by age brackets

Next, we examined the age distributions of derived subtypes for the two different age groups used in our study. For the UK Biobank participants, we categorized individuals into five age brackets based on their reported ages: 40–50, 50–55, 55–60, 60–65, 65–70 years. Similarly, for the ABCD Study participants, we grouped them into four groups: 100–110 months, 110–120 months, 120–130 months, 130–140 months. We then calculated the mean and standard deviation

of the brain scores for each age bracket, which can represent the central tendency and variability of chronotype-related brain patterns within each age group.

## Reporting summary

Further information on research design is available in the Nature Portfolio Reporting Summary linked to this article.

## Data availability

Data from the UK Biobank are available to all qualified researchers upon submission of an application. All instructions and procedures for access can be found on the UK Biobank website (https://www.ukbiobank.ac.uk/enable-your-research/apply-for-access). The ABCD Study data are administered by the National Institutes of Mental Health Data Archive and is freely available to qualified researchers who submit an access request. Instructions for obtaining the data are available online (https://nda.nih.gov/abcd/request-access). The Harvard–Oxford atlas, Probabilistic cerebellar atlas, and Johns Hopkins University atlas are accessible online (https://fsl.fmrib.ox.ac.uk/fsl/fslwiki/Atlases). Source data are provided with this paper.

## Code availability

The processing scripts for this work were implemented in Python (3.9.18) and utilized the following packages: scikit-learn (1.5.2), numpy (1.26.4), pandas (2.2.3), scipy (1.13.1), nilearn (0.11.0), mateplotlib (3.8.4), joblib (1.4.2), fMRIPrep 21.02, Nipype 1.6.1, ANTs 2.3.3, FSL 6.0.5.1, FreeSurfer 6.0.1. All related scripts are publicly accessible and archived at https://doi.org/10.5281/zenodo.17524317[130].

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

## Acknowledgements

We thank Professor Nicholas Christakis for reviewing an initial version of this manuscript and providing helpful comments. D.B. was supported by the Brain Canada Foundation through the Canada Brain Research Fund, with the financial support of Health Canada, the National Institutes of Health (NIH R01 AG068563A, NIH R01 DA053301-01A1 and NIH R01 MH129858-01A1), the Canadian Institute of Health Research (CIHR 438531 and CIHR 470425), the Healthy Brains Healthy Lives initiative (Canada First Research Excellence fund), Google (Research Award, Teaching Award), and by the CIFAR Artificial Intelligence Chairs programme (Canada Institute for Advanced Research). L.Z. was funded by the China Scholarship Council (CSC: 202106070134). The funders had no role in study design, data collection and analysis, decision to publish or preparation of the manuscript.

## Author contributions

D.B. and L.Z. conceived and executed the project and wrote the paper. K.S., J.M., S.A., J.K., J.C., K.-F.S. and R.I.M.D. contributed to the analysis and interpretation of the data, as well as revision of the paper. D.B. led data analytics.

## Competing interests

D.B. is a shareholder and advisory board member of MindState Design Labs, USA. The other authors declare no competing interests.
