## [Transparent Peer Review file · Nature Communications]

Latent brain subtypes of chronotype reveal unique behavioral and health profiles across cohorts

Corresponding Author: Professor Danilo Bzdok

Version 0:

Reviewer comments:

Reviewer #1

(Remarks to the Author)

Based on two large multimodal neuroimaging datasets, the authors used the classical PLS approach, from a data-driven perspective, to classify the chronotype into five subtypes, and further explore brain-behaviour association patterns. It is well written, and the authors discussed the findings in detail. Overall, I like this study and think it contributes to the current knowledge. However, I need a few things to be clarified.

The UK Biobank is for middle-aged and older adults, while the ABCD Study is for children. The circadian rhythms and sleep patterns of middle-aged and older adults and children are completely different. Please further explain the necessity and rationality for combining the two datasets.

The UK Biobank includes sMRI, DTI, and rs-fMRI, while the ABCD Study includes only sMRI. This concern is the interpretability and generalisation of the model.

The authors treated sleep-related variables (e.g., sleep duration, insomnia, snoring) as nuisance covariates and regressed them out from all brain imaging measures in downstream analyses. While this approach is statistically valid under the assumption that sleep factors are purely confounders, it may introduce theoretical and analytical biases if sleep behaviors lie on the causal pathway between chronotype and brain structure/function. Existing literature suggests that chronotype and sleep behaviors exhibit bidirectional relationships (e.g., Tussey EJ, Hillebrant-Openshaw M, Wong MM. Bidirectional relationships between chronotype and sleep hygiene in children with and without parental history of alcohol use disorder. *Sleep Health*. 2024 Dec.), and sleep disturbances may mediate—rather than merely confound—the association between chronotype and certain neurological outcomes (e.g., Wenzler AN, Liefbroer AC, Voshaar RCO, Smidt N. Chronotype as a potential risk factor for cognitive decline: The mediating role of sleep quality and health behaviours in a 10-year follow-up study. *J Prev Alzheimers Dis*. 2025 Jun.). By regressing out sleep variables, the analysis risks removing biologically meaningful variance, potentially masking true mechanistic pathways or distorting effect estimates. If I understand correctly, the following analysis can be made:

Conduct sensitivity analyses comparing models with and without adjustment for sleep variables to assess their influence on effect sizes.

Explore stratified or interaction analyses (e.g., examining chronotype-brain associations separately in individuals with/without insomnia) to evaluate whether sleep factors modify these relationships.

These additional analyses would significantly improve the robustness and interpretability of the findings while addressing potential theoretical limitations.

Some coefficients are relatively low (below $r=0.1$). This result should be treated with caution (Schober P, Boer C, Schwarte LA. Correlation Coefficients: Appropriate Use and Interpretation. *Anesth Analg*. 2018 May.). The authors should: (a) contextualize their effect sizes relative to methodological approaches and findings in related literature, and (b) provide a more nuanced interpretation of these results in their discussion, addressing the potential implications of such small effect sizes.

While self-reported questionnaire-based chronotype assessments are generally considered reliable, existing validation studies have predominantly used comprehensive multi-item instruments (e.g., MEQ, MCTQ). The current study's reliance on a single self-report item from UK Biobank raises significant concerns about measurement validity. 'objective actigraphy-

based measures provide more nuanced information on daily activity patterns and are sensitive to chronotype variability across different days.' I recommend that the authors either: (a) incorporate actigraphy data to validate their findings, or (b) thoroughly discuss this methodological limitation and its potential impact on result interpretation.

I note that there are actually four responses: "definitely a morning person", "more a morning than an evening person", "more an evening than a morning person", "definitely an evening person". Please provide further justification for the conversion to a binary variable. This may lose gradient information and weaken statistical power.

And a little question about subtype 2 "depression-associated eveningness". The naming of subtypes using clinical diagnosis is somewhat inappropriate.

Reviewer #2

(Remarks to the Author)

The authors challenge the traditional view of chronotype as a uniform, stable trait, arguing instead that this perspective overlooks significant heterogeneity in the neural and behavioral profiles of both early risers and night owls. Their study introduces an innovative method to detect biologically valid chronotype subtypes.

They propose that chronotypes can be meaningfully differentiated based on brain characteristics. Using a supervised learning framework applied to large-scale multimodal neuroimaging and behavioral data from 27,030 UK Biobank participants, the authors identify five distinct chronotype subtypes. These subtypes are derived from clustering brain features, and subsequently correlated with deep behavioral profiles, medical diagnoses, and drug prescriptions.

The study aims to go beyond simple cerebral characteristics by highlighting specific behavioral and health-related patterns associated with each subtype:

- Subtype 1: A night owl profile associated with emotional regulation and cognitive performance.
- Subtype 2: A night owl subtype linked to depression, smoking, and cardiovascular risk.
- Subtype 3: A morningness subtype characterized by lower substance use and fewer health issues.
- Subtype 4: A morningness subtype more prevalent among women associated with lower testosterone levels, menstruation disorders, and depression phenotypes
- Subtype 5: An eveningness subtype more prevalent among men linked to elevated cardiovascular risks and male-special associations

To assess external validity, the authors replicated their findings in an independent sample of 10,550 U.S. children from the ABCD Study[®] cohort. Interestingly, they found low similarity in behavioral and diagnostic outcomes across corresponding subtypes, suggesting that these profiles may not generalize easily across age groups or populations.

This original work is compelling and attempts to address longstanding uncertainties in the relationship between chronotype and health by identifying a substantial heterogeneity in brain-chronotype-behavior patterns, as well as their complex interactions with age and sex. The use of a supervised learning framework on very large-scale multimodal data (brain imaging, behavior, diagnosis, prescriptions) adds robustness to the findings.

However, several limitations warrant consideration.

First, while the study introduces a data-driven approach, the resulting subtypes largely align with existing findings in the literature: morning types tend to be women with healthier profiles and lower risk behaviors, while evening types are more likely to be men with higher rates of substance use, cardiovascular and metabolic diseases, and poorer mental health.

Second, the clinical identification of the five subtypes by practitioners appears challenging for two main reasons: (1) the subtypes are defined using complex brain imaging measures—such as gray matter volume, white matter tract integrity, and intrinsic functional connectivity—that are not accessible in routine clinical practice; and (2) the phenotypic (behavioral and health-related patterns) differences between the subtypes are relatively subtle, making them difficult to distinguish based on observable clinical features alone.

By example, all three evening-oriented subtypes are associated with increased cardiovascular risk or treatments related to cardiac function. These 3 subtypes are associated with smoking. Subtypes 1 and 2, are both linked to mental health issues (e.g., depression or emotional difficulties). The authors attempt to distinguish these two subtypes based on tobacco use, suggesting that smoking in Subtype 2 may drive both its mental health problems and cardiovascular risk. However, this distinction is not entirely convincing, as Subtype 1 is also associated with tobacco and cannabis use. More notably, Subtype 2 is characterized by reduced physical activity, which may offer a more plausible explanation for the greater health burden observed in this group. The clinical distinctions between subtypes, particularly among evening types, are not always clear or convincing, contrary to what is suggested in the abstract and title, or even in the discussion. Moreover, gender may act as a discriminatory factor; however, for subtypes 4 and 5, it is primarily noted in terms of prevalence—subtype 4 being more common in women, and subtype 5 in men. For instance, although women may still occur in subtype 5, the other phenotypes are clearly male-dominated. This prompts the question of how women within subtype 5 are identified.

Third, it is surprising that neither sleep duration nor circadian misalignment were associated with the five identified subtypes, given that both are well-established correlates of chronotype and are strongly linked to health outcomes. Incorporating these variables could have strengthened the analysis and potentially enhanced the clinical differentiation between subtypes. If these variables were included in the models, the authors should clarify why they do not appear among the defining

characteristics of the subtypes.

Finally, in the introduction, the authors suggest that identifying these subtypes could pave the way for targeted chronotherapeutic interventions. However, it would be valuable if they could elaborate on what specific interventions might be appropriate for each of the five subtypes.

Overall, this original study contributes meaningfully to the field by emphasizing the neural and behavioral diversity within chronotypes, but additional discussion is needed to clarify the clinical implications and practical applications of these findings.

Version 1:

Reviewer comments:

Reviewer #1

(Remarks to the Author)

The authors have effectively addressed all criticisms from the reviewers and made a successful revision.

Reviewer #2

(Remarks to the Author)

The authors have satisfactorily addressed most of my initial comments and have substantially improved the manuscript. The authors have addressed most of my previous comments and have considerably strengthened the manuscript. However, I would have appreciated a more explicit discussion in the revised text of the clinical relevance and potential translational implications of their findings.

For me, the main originality of this manuscript lies in demonstrating that trajectories of behaviors and disease emergence can diverge even within the same chronotype category, and that gender is one of the key variables shaping these trajectories. This multidimensional view of chronotype subtyping provides a valuable conceptual advance over traditional, monolithic approaches. This is not really explained in the text and summary.

That said, while the identified subtypes exhibit distinct brain-based markers, they remain defined at a single time point ("snapshot"). From a translational or clinical perspective, it would have been more informative to identify predictive biomarkers capable of anticipating subtype membership or transitions between subtypes over time. Such predictive markers would enable early detection and preventive interventions before maladaptive behaviors or pathological processes become established. Although the reported brain-based markers are intriguing and biologically consistent, they are likely difficult to identify or quantify in the general population with current neuroimaging tools. This limitation somewhat constrains the immediate clinical applicability of the proposed subtyping framework and calls for future work aimed at translating these neurobiological signatures into more accessible, behavioral, or physiological predictive markers.

Finally, the authors could strengthen the limitations section by acknowledging that the study lacks an objective marker of circadian phase (as noted by another reviewer) and does not include a measure of social jet lag...

We thank the reviewers and editors for their valuable feedback and the chance to resubmit our work. Our point-by-point responses to the comments of reviewers are detailed below. The manuscript has been revised accordingly, and we believe that the updated manuscript has been substantially improved.

REVIEWER COMMENTS

Reviewer #1 (Remarks to the Author):

Based on two large multimodal neuroimaging datasets, the authors used the classical PLS approach, from a data-driven perspective, to classify the chronotype into five subtypes, and further explore brain-behaviour association patterns. It is well written, and the authors discussed the findings in detail. Overall, I like this study and think it contributes to the current knowledge. However, I need a few things to be clarified.

We thank the reviewer's supportive comments.

The UK Biobank is for middle-aged and older adults, while the ABCD Study is for children. The circadian rhythms and sleep patterns of middle-aged and older adults and children are completely different. Please further explain the necessity and rationality for combining the two datasets.

We thank the reviewer for this insightful comment. Note that both datasets - UKB and ABCD - were analyzed in different steps of the analysis; that is, we did not combine the subjects for joint analysis at any step of the way.

We agree that circadian rhythms and sleep patterns differ significantly across the lifespan, as we discussed in the manuscript. To clarify, our goal was not to combine the two datasets to create a single model, which may obscure the important age-related differences as the reviewer indicated. Instead, we employed a cross-cohort validation framework to test the stability of the relationships between brain, behavior, and health obtained in UKB by extension to ABCD.

This approach would address a key question: are the chronotype patterns stable across the lifespan, or are they age-specific? If the middle-to-old age derived patterns generalized to the child cohort, it suggests a stable or perhaps fundamental association of these traits in the human brain-behavior association that exists from childhood through older adulthood (such as typical night owl subtype 1, and sex-related subtype 4 and 5). If it does not generalize, it provides compelling evidence that these brain-behavior relationships are age-specific (or cohort-specific), which is itself a valuable finding (such as the mood-related subtype 2). Overall, using the ABCD Study as a validation cohort

provides a powerful test of the robustness and developmental trajectory of our findings, significantly strengthening our findings and conclusions.

To incorporate this important point and enhance our manuscript, we have added more specific rationale in our manuscript: “The application of our adult-derived model to the child cohort allowed us to probe whether the observed brain-behavior relationships are stable across the lifespan or emerge specifically in the older adulthood.” (Line 865-867)

The UK Biobank includes sMRI, DTI, and rs-fMRI, while the ABCD Study includes only sMRI. This concern is the interpretability and generalisation of the model.

We thank the reviewer for this important point regarding generalizability. Given the unique richness and depth of the UK Biobank dataset, it is challenging to find an external cohort with an identical level of multi-modal data for validation. We agree that the absence of DTI and rs-fMRI data in the ABCD Study means we could only validate the sMRI-related information of our brain-chronotype patterns. We have openly acknowledged this limitation in our manuscript: “Notably, validation based solely on sMRI in the ABCD Study cohort would reduce the power of the holistic model derived from multi-modal data. Despite this likely loss in power, we validated the majority of our UKB-derived chronotypes in the ABCD study.” (line 327-330)

In particular, the partial validation we conducted provides critical and nuanced insights. Our results show that the significant associations, such as the link between high cognitive ability and night owls (subtype 1), as well as the sex-associated subtypes (subtype 4 and 5), were replicated in the ABCD cohort. This suggests that these specific patterns have strong anatomical correlates that are detectable even with a single data modality. Furthermore, the age-related distribution (Fig. 7) aligns with established life-span trends (Foster & Roenneberg, 2008; Duffy et al., 2015; Duffy et al., 2017), lending additional credibility to our findings in UKB participants. On the other hand, the incomplete generalization of other aspects (such as mood-related subtype 2) implies that some pattern elements may be rooted in other modalities and require multi-modal assessment. In conclusion, while multi-modal data is ideal for a holistic view, the successful replication of partial core findings using only the sMRI still supports the generalizability of the main patterns. We have discussed this limitation and agree that future research would gain added value from multi-modal validation across cohorts.

The authors treated sleep-related variables (e.g., sleep duration, insomnia, snoring) as nuisance covariates and regressed them out from all brain imaging measures in downstream analyses. While this approach is statistically valid under the assumption that sleep factors are purely confounders, it may introduce theoretical and analytical biases if sleep behaviors lie on the causal pathway between chronotype and brain

structure/function. Existing literature suggests that chronotype and sleep behaviors exhibit bidirectional relationships (e.g., Tussey EJ, Hillebrant-Openshaw M, Wong MM. Bidirectional relationships between chronotype and sleep hygiene in children with and without parental history of alcohol use disorder. Sleep Health. 2024 Dec.), and sleep disturbances may mediate—rather than merely confound—the association between chronotype and certain neurological outcomes (e.g., Wenzler AN, Liefbroer AC, Voshaar RCO, Smidt N. Chronotype as a potential risk factor for cognitive decline: The mediating role of sleep quality and health behaviours in a 10-year follow-up study. J Prev Alzheimers Dis. 2025 Jun.). By regressing out sleep variables, the analysis risks removing biologically meaningful variance, potentially masking true mechanistic pathways or distorting effect estimates. If I understand correctly, the following analysis can be made:

Conduct sensitivity analyses comparing models with and without adjustment for sleep variables to assess their influence on effect sizes.

Explore stratified or interaction analyses (e.g., examining chronotype-brain associations separately in individuals with/without insomnia) to evaluate whether sleep factors modify these relationships.

These additional analyses would significantly improve the robustness and interpretability of the findings while addressing potential theoretical limitations.

We thank the reviewer for these constructive comments and suggestions. We have conducted the suggested sensitivity analyses by comparing our primary model (which regressed out sleep duration, insomnia, snoring, dozing, napping) with a model that did not adjust for these sleep variables. The two models differed only in whether they regressed out the sleep phenotypes from brain imaging data, all other steps were the same. The results demonstrated that the derived brain scores and loadings were highly correlated between two models (all Pearson's $r > 0.9$, $P < 0.001$, see below and supplementary Fig. 7). This indicates that the exclusion and inclusion of sleep covariates had a negligible impact on the core patterns identified by our main model and hence on our scientific conclusions.

We have updated our manuscript: “We also conducted sensitivity analyses by comparing models with and without regressing out sleep phenotypes. The derived scores and loadings were significantly correlated between two models (all Pearson's $r > 0.9$, $P < 0.001$, see Supplementary Fig. 7), indicating negligible effects of sleep phenotypes on our main modeling.” (line 668-671)

Supplementary Fig. 7 | Sensitivity Analyses assessing the impact of sleep phenotypes on our main modeling. Comparison of brain scores (a) and brain loadings (b) from models with versus without adjustment for sleep phenotypes (sleep duration, insomnia, dozing, snoring, napping). Components were aligned to the five significant subtypes from the primary analysis. Pearson correlations for all subtypes were > 0.9 ($P < 0.001$), indicating that adjusting for sleep phenotypes had a negligible effect on the derived patterns.

Some coefficients are relatively low (below $r=0.1$). This result should be treated with caution (Schober P, Boer C, Schwarte LA. Correlation Coefficients: Appropriate Use and Interpretation. Anesth Analg. 2018 May.). The authors should: (a) contextualize their effect sizes relative to methodological approaches and findings in related literature, and (b) provide a more nuanced interpretation of these results in their discussion, addressing the potential implications of such small effect sizes.

We thank the reviewer for this helpful point. We agree that small effect sizes, while statistically significant, require careful contextualization. We also recognized that using Pearson's correlation between brain scores and binary chronotype scale may not be an appropriate approach. Thus, we have replaced this analysis with two-sample t-tests to compare the morningness and eveningness groups for each subtype. Specifically, for each subtype, derived brain scores were grouped by participant chronotype (morningness or eveningness). The two-sample t-tests were then conducted to test that brain scores are greater in the morningness group. The results robustly support our conclusions: subtype 1 indicating greater eveningness ($t = -16.94, P < 0.001$), subtype 2 indicating greater eveningness ($t = -15.81, P < 0.001$), subtype 3 indicating greater morningness ($t = 8.44, P < 0.001$), subtype 4 indicating greater morningness ($t = 6.74, P < 0.001$), subtype 5 indicating greater eveningness ($t = -2.79, P < 0.01$). We have replaced Pearson's

correlation by two-sample t-test results in our manuscript. (line 133-135, line 180-181, line 216-217, line 253-256, line 282-286)

More broadly, the effect sizes on pheno-wide association study (PheWAS) on biobank population scale studies were often small yet statistically significant and biologically meaningful. This is a well-established characteristic of the population imaging field, as highlighted by the seminal paper published in Nature Neuroscience by the creators of the UK Biobank cohort themselves as they carried out phenome-wide association analyses almost 10 years ago (Miller et al., 2016), and supported by our previous work (Saltoun et al., 2023, Zhou et al., 2025). The strength of our approach lies not in the magnitude of a single correlation, but in the consistent, multivariate pattern identified across thousands of variables, which provides a holistic and stable view of chronotypes. All statistically significant results were provided in the supplementary materials, which contains more nuanced information.

Miller, K. L., et al. (2016). "Multimodal population brain imaging in the UK Biobank prospective epidemiological study." Nature Neuroscience.

Saltoun, K., et al. (2023). "Dissociable brain structural asymmetry patterns reveal unique phenome-wide profiles." Nature Human Behaviour.

Zhou, L., et al. (2025). "Multimodal population study reveals the neurobiological underpinnings of chronotype." Nature Human Behaviour.

*While self-reported questionnaire-based chronotype assessments are generally considered reliable, existing validation studies have predominantly used comprehensive multi-item instruments (e.g., MEQ, MCTQ). The current study's reliance on a single self-report item from UK Biobank raises significant concerns about measurement validity. 'objective actigraphy-based measures provide more nuanced information on daily activity patterns and are sensitive to chronotype variability across different days.' I recommend that the authors either: (a) incorporate actigraphy data to validate their findings, or (b) **thoroughly discuss this methodological limitation and its potential impact on result interpretation.***

We thank the reviewer for the important methodological point. The single question used by UK Biobank (MEQ question 19) has been directly validated against the full MEQ showing a strong correlation ($r = 0.703$) (Leocadio-Miguel et al., 2021). The self-reported measures are also known to be significantly associated with objective actigraphy assessments (Gershon et al., 2018, Schneider et al., 2022). Furthermore, a genetic analysis within the UK Biobank demonstrated high concordance between self-reported and actigraphy-derived chronotype measurements (Jones, et al., 2019). Collectively, above evidence establishes that the UK Biobank single-item chronotype measure is valid

and captures meaningful biological information across behavior, body movement patterns and genetic profile.

Moreover, to further enhance robustness, we excluded participants with inconsistent chronotype reports between their initial assessment and the imaging visit, ensuring measurement stability (see Methods).

We totally agree with the necessity of thoroughly discussing this methodological limitation, which is why we have updated the discussion as suggested by this reviewer: “While questionnaire-based measurements of chronotype have been shown to be reliable and robust (Leocadio-Miguel et al., 2021, Gershon et al., 2018, Jones, et al., 2019, Schneider et al., 2022), subjective factors may introduce potential confounding that could contaminate our findings. Future investigations could benefit from more directly incorporating objective measures, such as actigraphy data from wearable devices, mid-sleep time or blood melatonin fluctuations, to capture a more nuanced and dynamic representation of chronotype.” (line 558-563)

Leocadio-Miguel, M. A., et al. (2021). "Compared Heritability of Chronotype Instruments in a Single Population Sample." *Journal of Biological Rhythms*.

Gershon, A., et al. (2018). "Subjective versus objective evening chronotypes in bipolar disorder." *Journal of Affective Disorders*.

Schneider, J., et al. (2022). "Human chronotype: Comparison of questionnaires and wrist-worn actigraphy." *Chronobiology International*.

Jones, S. E., et al. (2019). "Genome-wide association analyses of chronotype in 697,828 individuals provides insights into circadian rhythms." *Nature Communications*.

*I note that there are actually four responses: "definitely a morning person", "more a morning than an evening person", "more an evening than a morning person", "definitely an evening person". **Please provide further justification for the conversion to a binary variable. This may lose gradient information and weaken statistical power.***

We thank the reviewer for raising this important point. Our study was built on previous genetic study of chronotype in the UK Biobank (Jones, et al., *Nature Communications*, 2019). The authors of this previous study employed the same binary coding (morningness/cases: definitely a morning person, more a morning than an evening person; eveningness/controls: definitely an evening person, more an evening person than a morning person) on UK Biobank dataset to enhance the interpretability of each allele. This approach also enabled harmonization with the 23andMe cohort, which used a binary chronotype definition. This genetic study discovered several significant genome-wide locations enriched for circadian regulation genes and further demonstrated that the

morning chronotype was genetically negatively correlated with depressive symptoms, major depressive disorder, and intelligence.

Moreover, comparison with actigraphy-derived sleep measurements in their study showed that self-reported chronotype loci were related to sleep timing rather than sleep duration or fragmentation, further supporting the validity of the UK Biobank self-report chronotype assessment and its biological relevance.

Additionally, because our study excluded participants with inconsistent chronotype reports across two time points, using a binary classification is more tolerant of minor fluctuations within the morningness and eveningness categories. This allowed us to retain a larger sample size, thereby preserving statistical power while ensuring a stable chronotype classification for our analyzed participant sample.

We have also conducted a sensitivity analysis by comparing the 4-scale assessments and binary assessments model. After aligning the 4-scale model to our five significant subtypes, we found that both brain scores and loadings were highly correlated between two models (all Pearson's $r > 0.9$, $P < 0.001$) (see below and supplementary Fig. 8). This suggested that the core multivariate patterns we identified are robust and not meaningfully altered by the binary encoding of our target chronotype phenotype.

We have updated our manuscript: **“Moreover, we conducted sensitivity analyses by comparing models with binary and the raw 4-level encoding of the chronotype variable. The derived scores and loadings were significantly correlated between two models (all Pearson's $r > 0.9$, $P < 0.001$, see Supplementary Fig. 8), indicating the core multivariate patterns derived in our main analysis are robust and not meaningfully altered by our binary target phenotype.”** (line 621-625)

Supplementary Fig. 8 | Sensitivity analyses assessing the impact of binary chronotype on our main modeling. Comparison of brain scores (a) and brain loadings (b) from models with 4-scale versus binary scale of chronotype. Components were aligned to the five significant subtypes from the primary analysis. Pearson correlations for

all subtypes were > 0.9 ($P < 0.001$), indicating that binary simplification of chronotype had a negligible effect on the derived patterns.

And a little question about subtype 2 “depression-associated eveningness”. The naming of subtypes using clinical diagnosis is somewhat inappropriate.

We thank the reviewer for their comments. We have updated the naming of subtype 2 to “mood-associated eveningness” in the manuscript. (line 33)

Reviewer #2 (Remarks to the Author):

The authors challenge the traditional view of chronotype as a uniform, stable trait, arguing instead that this perspective overlooks significant heterogeneity in the neural and behavioral profiles of both early risers and night owls. Their study introduces an innovative method to detect biologically valid chronotype subtypes.

They propose that chronotypes can be meaningfully differentiated based on brain characteristics. Using a supervised learning framework applied to large-scale multimodal neuroimaging and behavioral data from 27,030 UK Biobank participants, the authors identify five distinct chronotype subtypes. These subtypes are derived from clustering brain features, and subsequently correlated with deep behavioral profiles, medical diagnoses, and drug prescriptions.

The study aims to go beyond simple cerebral characteristics by highlighting specific behavioral and health-related patterns associated with each subtype:

- *Subtype 1: A night owl profile associated with emotional regulation and cognitive performance.*
- *Subtype 2: A night owl subtype linked to depression, smoking, and cardiovascular risk.*
- *Subtype 3: A morningness subtype characterized by lower substance use and fewer health issues.*
- *Subtype 4: A morningness subtype more prevalent among women associated with lower testosterone levels, menstruation disorders, and depression phenotypes*
- *Subtype 5: An eveningness subtype more prevalent among men linked to elevated cardiovascular risks and male-special associations*

To assess external validity, the authors replicated their findings in an independent sample of 10,550 U.S. children from the ABCD Study® cohort. Interestingly, they found low similarity in behavioral and diagnostic outcomes across corresponding subtypes,

suggesting that these profiles may not generalize easily across age groups or populations.

This original work is compelling and attempts to address longstanding uncertainties in the relationship between chronotype and health by identifying a substantial heterogeneity in brain-chronotype-behavior patterns, as well as their complex interactions with age and sex. The use of a supervised learning framework on very large-scale multimodal data (brain imaging, behavior, diagnosis, prescriptions) adds robustness to the findings.

We thank the reviewer for their enthusiastic feedback.

However, several limitations warrant consideration.

First, while the study introduces a data-driven approach, the resulting subtypes largely align with existing findings in the literature: morning types tend to be women with healthier profiles and lower risk behaviors, while evening types are more likely to be men with higher rates of substance use, cardiovascular and metabolic diseases, and poorer mental health.

We thank the reviewer for this insightful observation. We agree that the broad associations we find (e.g., morning types with healthier lifestyle profiles, evening types with higher-risk behaviors) are consistent with the established literature. This consistency, in fact, serves as a strong validation of our data-driven methodology, confirming that it captures biologically and behaviorally meaningful patterns.

The primary novelty of our analytical approach lies in moving beyond these two broad black-and-white categories by disentangling the significant heterogeneity within them. As one of many possible examples, previous findings on topics like the chronotype-depression link have been mixed (Norbury, 2021). Our approach provides a clear explanation for this inconsistency by revealing distinct subtypes, such as the typical night owl subtype 1 without depressed mood, while another night owl subtype 2 related to depressed mood. This suggests that what is often grouped together as “eveningness” actually comprises distinct subgroups with different risk profiles.

Overall, our study refinement of these categories into more homogeneous subtypes that we show and validate to be robust at population scale, offering a clearer understanding of the specific associations within the morningness and eveningness populations. This nuanced view helps resolve conflicting findings in the literature.

Second, the clinical identification of the five subtypes by practitioners appears challenging for two main reasons: (1) the subtypes are defined using complex brain imaging measures—such as gray matter volume, white matter tract integrity, and intrinsic

functional connectivity—that are not accessible in routine clinical practice; and (2) the phenotypic (behavioral and health-related patterns) differences between the subtypes are relatively subtle, making them difficult to distinguish based on observable clinical features alone.

By example, all three evening-oriented subtypes are associated with increased cardiovascular risk or treatments related to cardiac function. These 3 subtypes are associated with smoking. Subtypes 1 and 2, are both linked to mental health issues (e.g., depression or emotional difficulties). The authors attempt to distinguish these two subtypes based on tobacco use, suggesting that smoking in Subtype 2 may drive both its mental health problems and cardiovascular risk. However, this distinction is not entirely convincing, as Subtype 1 is also associated with tobacco and cannabis use. More notably, Subtype 2 is characterized by reduced physical activity, which may offer a more plausible explanation for the greater health burden observed in this group. The clinical distinctions between subtypes, particularly among evening types, are not always clear or convincing, contrary to what is suggested in the abstract and title, or even in the discussion. Moreover, gender may act as a discriminatory factor; however, for subtypes 4 and 5, it is primarily noted in terms of prevalence—subtype 4 being more common in women, and subtype 5 in men. For instance, although women may still occur in subtype 5, the other phenotypes are clearly male-dominated. This prompts the question of how women within subtype 5 are identified.

We thank the reviewer for this critical and insightful point. We totally agree that the brain-imaging derived measurements are not readily accessible in routine clinical practice. To clarify, our data-driven approach does not aim to create discrete, mutually exclusive diagnostic boxes akin to medical diagnoses. Instead, it identifies latent neurobiological dimensions of chronotypes that exist in the population - each of which exists in a given person in continuous degrees.

A helpful analogy to our 5 chronotype model is the “Big five model” of personality, with constructs like “conscientiousness” or “neuroticism” because of their strong predictive power for a suite of real-world life outcomes, even though they are latent traits inferred from behavior. Similarly, our five subtypes of chronotype could serve as five dimensions. Each individual can be profiled by their scores on each dimension, with their dominant subtype simply being the dimension on which they score highest.

The dimensional framework could explain the phenotypic overlaps noted by the reviewer. The night owl subtypes (subtype 1, 2, 5) are expected to share some risks (such as the cardiovascular risks), but are distinguished by their unique combinations of other traits. We thank the reviewer for pointing out the key role of reduced physical activity in subtype 2, we have emphasized it in our manuscript (line 210-211, Fig. 2 caption).

Furthermore, in the dimensional model, a woman assigned to subtype 5 means her overall multivariate pattern, across all brain, behavioral, and health measures, aligns most

strongly with the male-dominated subtype 5 profile. Her sex is one point in that pattern, but her combination of other traits overrides the typical sex association, making her identification within this subgroup statistically and biologically informative.

To incorporate insights of this and below points, we have updated discussion in our manuscript: “Notably, our data-driven approach aims not to create a discrete diagnostic clinical tool. Instead, we identified latent neurobiological dimensions of chronotype that exist in the population, which could pave the way for personalized interventions by identifying specific risk profiles and biological pathways for future validation in dedicated interventional or animal studies. Chronotypes themselves are not disorders that require treatments, but some disorders may relate to certain components of chronotype, their practical interventions may also benefit from our framework. For instance, the distinct depression-associated profiles we identified suggest different therapeutic strategies. A depressed patient with the mood-associated eveningness subtype 2 profile might respond best to an integrated approach combining circadian realignment (e.g., bright light therapy, Chan et al., 2022) with physical activity protocols. Conversely, a depressed patient with morningness subtype 4 profile, which was linked to female-related factors, reduced social engagement, and sexual assault, might benefit more from interventions focused on enhancing social support and connectivity. Furthermore, future studies specifically designed to capture circadian misalignment will be a valuable next step for refining these profiles for clinical applications.” (line 572-586)

Chan, J. W. Y., et al. (2022). "Change in circadian preference predicts sustained treatment outcomes in patients with unipolar depression and evening preference." *Journal of clinical sleep medicine: JCSM: official publication of the American Academy of Sleep Medicine*.

Third, it is surprising that neither sleep duration nor circadian misalignment were associated with the five identified subtypes, given that both are well-established correlates of chronotype and are strongly linked to health outcomes. Incorporating these variables could have strengthened the analysis and potentially enhanced the clinical differentiation between subtypes. If these variables were included in the models, the authors should clarify why they do not appear among the defining characteristics of the subtypes.

We thank the reviewer for this important observation. We would like to clarify that the UK Biobank does not contain a direct measure of circadian misalignment. Therefore, including this construct in our quantitative analyses was not feasible for our primary model. To mitigate potential confounding from circadian disruption, we excluded

participants with shift-work experience and retained only those with stable self-reported chronotypes across two assessment time points (see Methods).

Furthermore, the absence of sleep phenotypes (including sleep duration) was an intentional consequence of our analytical design. Our goal was to isolate the neurobiological correlates of chronotype that are independent of sleep per se. To achieve this, we regressed out sleep phenotypes (sleep duration, insomnia, snoring, dozing, napping) from the brain imaging data before all downstream analyses (see Methods). Thus, the derived subtypes are defined by the variance in brain features that is not explained by these sleep phenotypes. Moreover, our sensitivity analysis confirmed that regressing out sleep variables did not distort the core findings, as the subtypes derived with and without sleep regression were nearly identical (all Pearson's r in brain scores and brain loadings were greater than 0.9, Supplementary Fig. 7).

In summary, our approach was designed to mitigate confounding related to sleep duration and circadian misalignment. We agree future studies specifically designed to capture circadian misalignment will be a valuable next step. We have updated our discussion (see above point).

Finally, in the introduction, the authors suggest that identifying these subtypes could pave the way for targeted chronotherapeutic interventions. However, it would be valuable if they could elaborate on what specific interventions might be appropriate for each of the five subtypes.

We thank the reviewer for this valuable comment and suggestion. To clarify, our intention is not to frame chronotypes as disorders requiring treatment, but rather to suggest that specific chronotype components may be associated with certain disease profiles. Identifying these components could inform more personalized intervention strategies.

For instance, the strong association between subtype 2 and depression suggests that depressed patients with this mood-associated eveningness profile may respond best to interventions that integrate circadian realignment (e.g. bright light therapy, Chan et al., 2022) with physical activity protocols. Conversely, a depressed patient with subtype 4 profile, which was also linked to depression but along with female-related factors (such as the menstruation disorders), reduced social engagement, and sexual assault, might benefit more from interventions enhancing social support and connectivity.

Overall, our framework of subtypes is not a ready-made clinical tool, but it paves the way for targeted interventions by identifying specific biological pathways and risk profiles that future, dedicated interventional or animal studies can target. We have enhanced the discussion section in our manuscript (see above point).

Overall, this original study contributes meaningfully to the field by emphasizing the neural and behavioral diversity within chronotypes, but additional discussion is needed to clarify the clinical implications and practical applications of these findings.

We thank the reviewers for the above insightful comments again. We have incorporated these important points - this reviewer feedback has therefore much enhanced our manuscript.

REVIEWERS' COMMENTS

Reviewer #1 (Remarks to the Author):

The authors have effectively addressed all criticisms from the reviewers and made a successful revision.

We thank the reviewer again for their insightful comments and suggestions which have substantially improved our manuscript.

Reviewer #2 (Remarks to the Author):

The authors have satisfactorily addressed most of my initial comments and have substantially improved the manuscript. The authors have addressed most of my previous comments and have considerably strengthened the manuscript. However, I would have appreciated a more explicit discussion in the revised text of the clinical relevance and potential translational implications of their findings.

For me, the main originality of this manuscript lies in demonstrating that trajectories of behaviors and disease emergence can diverge even within the same chronotype category, and that gender is one of the key variables shaping these trajectories. This multidimensional view of chronotype subtyping provides a valuable conceptual advance over traditional, monolithic approaches. This is not really explained in the text and summary.

That said, while the identified subtypes exhibit distinct brain-based markers, they remain defined at a single time point (“snapshot”). From a translational or clinical perspective, it would have been more informative to identify predictive biomarkers capable of anticipating subtype membership or transitions between subtypes over time. Such predictive markers would enable early detection and preventive interventions before maladaptive behaviors or pathological processes become established. Although the reported brain-based markers are intriguing and biologically consistent, they are likely difficult to identify or quantify in the general population with current neuroimaging tools. This limitation somewhat constrains the immediate clinical applicability of the proposed subtyping framework and calls for future work aimed at translating these neurobiological signatures into more accessible, behavioral, or physiological predictive markers. Finally, the authors could strengthen the limitations section by acknowledging that the study lacks an objective marker of circadian phase (as noted by another reviewer) and does not include a measure of social jet lag...

We thank the reviewer for this encouraging and thoughtful comment. We have now expanded our discussion section to explicitly discuss the clinical and translational

relevance of our findings. Specifically, we have added text: “In addition, our cross-sectional approach captured a snapshot rather than the temporal dynamics of these subtypes. Future longitudinal studies will be valuable for revealing transitions and dynamic changes both between and within subtypes, which would further enable early detection and preventive interventions before maladaptive behaviors or pathological processes become established.”

We have also discussed the objective measures of chronotype and social jet lag in our discussion section: “While questionnaire-based measurements of chronotype have been shown to be reliable and robust, subjective factors may introduce potential confounding that could contaminate our findings. Future investigations could benefit from more directly incorporating objective measures, such as actigraphy data from wearable devices, mid-sleep time or blood melatonin fluctuations, to capture a more nuanced and dynamic representation of chronotype.”

“Furthermore, to avoid potential confounds arising from circadian misalignment (social jet lag), our study excluded participants with a history of shift work. Future studies specifically designed to capture circadian misalignment will be a valuable next step for refining these profiles in this specific population and for advancing future clinical applications.”

We sincerely appreciate all the constructive comments and suggestions from the reviewers and the editorial team.